# Germline mutation rates and fine-scale recombination parameters in zebra finch

Djivan Prentout[1]*, Daria Bykova[1], Carla Hoge[1¤a], Daniel M. Hooper[2], Callum S. McDiarmid[3], Felix Wu[4¤b], Simon C. Griffith[3], Marc de Manuel[1¤c‡]*, Molly Przeworski[1,4‡]*

**1** Department of Biological Sciences, Columbia University, New York, New York, United States of America, **2** Institute for Comparative Genomics and Richard Gilder Graduate School, American Museum of Natural History, New York, New York, United States of America, **3** School of Natural Sciences, Macquarie University, Sydney, New South Wales, Australia, **4** Department of Systems Biology, Columbia University, New York, New York, United States of America

‡ Joint senior authors.
¤a Current address: Department of Human Genetics, University of Chicago, Chicago, United States of America
¤b Current address: Department of Organismal and Evolutionary Biology, Harvard University, Cambridge, United States of America
¤c Current address: Institute of Evolutionary Biology (CSIC-Universitat Pompeu Fabra), Barcelona, Spain
* djivanprentout11@gmail.com (DP); marc.demanuel@ibe.upf-csic.es (MdM); mp3284@columbia.edu (MP)

## Abstract

Most of our understanding of the fundamental processes of mutation and recombination stems from a handful of disparate model organisms and pedigree studies of mammals, with little known about other vertebrates. To gain a broader comparative perspective, we focused on the zebra finch (*Taeniopygia castanotis*), which, like other birds, differs from mammals in its karyotype (which includes many micro-chromosomes), in the mechanism by which recombination is directed to the genome, and in aspects of onto-genesis. We collected genome sequences from three generation pedigrees that provide information about 80 meioses, inferring 202 single-point *de novo* mutations, 1,088 crossovers, and 275 non-crossovers. On that basis, we estimated a sex-averaged mutation rate of $5.0 \times 10^{-9}$ per base pair per generation, on par with mammals that have a similar generation time (~2–3 years). Also as in mammals, we found a paternal germline mutation bias at later stages of gametogenesis (of 1.7:1) but no discernible difference between sexes in early development. Examining recombination patterns, we found that the sex-averaged crossover rate on macro-chromosomes is 0.93 cM/Mb, with a pronounced enrichment of crossovers near telomeres. In contrast, non-crossover rates are more uniformly distributed. On micro-chromosomes, sex-averaged crossover rates are substantially higher (3.96 cM/Mb), in accordance with crossover homeostasis, and both crossover and non-crossover events are more uniformly distributed. At a finer scale, recombination events overlap CpG islands more often than expected by chance, as expected in the absence of PRDM9. Estimates of the degree of GC-biased gene conversion (59%), the mean non-crossover conversion tract length (~32 bp), and the

**Data availability statement:** The whole genome sequencing data and the bisulfite sequencing data are available in the GenBank Sequence Read Archive under the accession numbers PRJNA1152924 and PRJNA1152957, respectively. The code is available at https://doi.org/10.5281/zenodo.13696268. Numerical data that underlies graphs or summary statistics are provided in S1 and S2 Data files.

**Funding:** The zebra finches were established in captivity and maintained with support from the Australian Research Council (FT130101253 to SCG). This work was supported by the National Institutes of Health (R01 GM83098 to MP; R35 GM153355 to MP), and a Human Frontiers Science Program Fellowship (LT000257/2021-L to MM). The funders had no role in study design, data collection and analysis, decision to publish, or preparation of the manuscript.

**Competing interests:** The authors have declared that no competing interests exist.

non-crossover-to-crossover ratio (5.4:1) are all comparable to those reported in primates and mice. Therefore, properties of germline mutation and recombination resolutions remain similar over large phylogenetic distances.

## Author summary

Germline mutation and meiotic recombination are fundamental genetic processes that give rise to genetic diversity and fuel evolution. Most of our knowledge about these processes stems from a handful of model organisms and studies in mammals. Here, we characterized properties of mutation and recombination in the zebra finch (*Taeniopygia castanotis*), which, like other birds, differs from mammals in several potentially salient respects, including its karyotype and the mechanism by which recombination is directed to the genome. By sequencing the genomes of three-generation pedigrees, we identified *de novo* mutations and recombination events (both crossovers and non-crossovers). Our analysis indicates that the sex-averaged mutation rate is comparable to those of mammals with similar generation times. Several aspects of recombination resolutions are also similar to those in mammals, notably the estimated ratio of crossovers to non-crossovers. Thus, our findings indicate that many aspects of mutation and recombination remain conserved over large phylogenetic distances.

## Introduction

Germline mutation and meiotic recombination are fundamental biological processes and the sources of heritable variation. The rates at which they occur are key parameters in evolutionary models and enable phylogenetic dating. Yet their properties have been studied in depth in only a handful of model organisms, primarily yeast species, mice, *Arabidopsis thaliana*, and a couple of Drosophila species (e.g., [1–6]). More recently, such studies have been complemented by sequencing pedigrees, primarily of mammals (e.g., [6–10]), as well as by *high-fidelity* (HiFi) long-read sequencing of male germ cells of humans and other primates [11–13].

Intriguingly, mutation parameters appear to be quite stable across mammalian species: for instance, the mutation rate per base pair (bp) per generation in mice (~0.5 x $10^{-8}$; [14–16]) is only around two times lower than that in humans ($1.2 \times 10^{-8}$; e.g., [17,18]) despite their generation time being around 50 times shorter [15]. In the soma, in turn, the mutation burden in colonic crypts appears relatively constant across 16 mammalian species at their typical lifespan (i.e., on average, a mammalian colonic epithelial cell carries around 3,000 *de novo* mutations at the time of death) [19]. Given that modifiers of mutation are known to segregate in mammals [20–22], the stability of mutation rates suggests they are conserved by natural selection [19,23]. It remains an open question, however, whether this conservation extends to broader phylogenetic distances.

Birds offer an interesting comparison to mammals in this regard, as the two taxa diverged approximately 320 million years ago [24] and differ in many potentially salient

features. For one, most passerines are seasonal breeders with spermatogenesis occurring during the breeding season, unlike many mammals (e.g., mice and humans), which typically produce sperm continuously [25,26]. Moreover, sex differences may appear earlier in bird ontogenesis, given that the avian sexual phenotype is directly determined by the sex chromosome content of individual cells [27,28], consistent with reported sex differences in primordial germ cell phenotypes before gonadal development [29].

To date, germline mutation rates have been estimated from pedigree studies in the collared flycatcher [30], the great reed warbler [31], as well as 18 other species [32], but inferences remain limited by the small number of trios considered per species (from one to eight, with a median of one trio). Most recently, an estimate was obtained in the zebra finch from genome sequences in 16 trios [33]. The authors report a mutation rate per generation in line with estimates in other passerines (~6 x $10^{-9}$ per base pair) [32], as well as a paternal mutation bias of ~4:1; a surprising feature was a relatively high transition:transversion ratio (3.4) compared to other vertebrates (2.3, on average across 151 species [32]). When the paternal bias arises during avian development remains unknown. It is also unclear if the mutational processes dominant in the mammalian germline [18,34,35] are active in the bird germline.

Birds further provide an interesting comparison to mammals with regard to recombination patterns. At a broad-scale, a number of salient features differ between the taxa: notably, avian genomes often harbor a large number of micro-chromosomes, which have higher average crossover rates—potentially up to 10-fold higher [36–38], replicate earlier and have a higher repeat content [39,40]. Moreover, birds lack the gene PRDM9 [41], which encodes the protein that directs recombination to the genome in most mammals (e.g., [42–44]), as well as in snakes and salmonids [45–47]. In rodent knockouts for PRDM9 and dogs that carry a pseudogene for PRDM9, recombination preferentially occurs at promoter-like features, in particular CpG islands [48–50]. Analyses of patterns of linkage disequilibrium (LD) suggest that in birds, population recombination rates are also elevated near CpG islands [36,51,52]. However, LD-based estimates can be biased by misspecification of the demographic model and by effects of selection, and mostly reflect the effects of crossovers rather than non-crossover events [53,54]. To our knowledge, only two studies have examined recombination events in birds more directly, based on genome sequences from pedigrees (in great reed warbler and flycatcher; [55,56]) and only in one case were non-crossovers identified in addition to crossovers [55]. Therefore, it remains unclear the extent to which both types of recombination events are concentrated at promoter-like features.

Other properties of recombination also remain to be studied in birds. In mammals, the non-crossover mean conversion tract length appears to be conserved between mice and primates (<100 bps; [6,8,11,13]). Yet this length is substantially shorter than that in *Drosophila melanogaster* or in yeast (*Saccharomyces cerevisiae*), which are estimated to be ~500 bps and 1.8 kb, respectively [2,3]. In turn, estimates of magnitude of the GC-biased gene conversion in mammals vary between 57% and 68% [6–9]. The extent of bias may depend on nucleotide diversity levels, as a recent study reported that only non-crossovers with a single mismatch in the gene conversion tract experience a transmission bias in mice and potentially humans [6]. In collared flycatcher, in which heterozygosity is ~0.4% [57], the authors reported a point estimate of 59% based on 229 non-crossover events [55]. Whether the same is true in a species with higher diversity remains to be tested. More generally, key aspects of meiotic recombination remain poorly characterized in birds, including the number and the distribution of non-crossover events as well as parameters such as the mean gene conversion tract length, the total number of double strand breaks (DSBs) or the non-crossover-to-crossover ratio.

To fill these gaps in our understanding and characterize fundamental mutation and recombination parameters in birds, we focused on the zebra finch, a well-studied passerine with a contiguous and high-quality genome assembly [58] and high levels of nucleotide diversity (~1%) [59]. To this end, we sequenced genomes of extended pedigrees to identify *de novo* point mutations (DNMs), as well as infer crossovers and non-crossovers.

## Results

### Germline *de novo* mutation

**Identification of sex-specific *de novo* mutations (DNMs) from pedigrees.** We generated whole genome sequences, on average at 25-fold coverage, from 74 zebra finch individuals in four three-generation pedigrees and four

trios, comprising 40 trios in total (Fig 1A and S1 Table). After mapping the reads of each individual to the zebra finch reference genome (RefSeq ID: GCF_003957565.2), we identified autosomal regions in the assembly to which short-read sequencing data could be reliably mapped and where the three individuals in the focal trio had sufficient but not unusually high depth of coverage (see Methods). This approach led us to retain an average of 502.8 Mb per trio (min = 475.3, max = 511.1), or 52% of the autosomal genome. We then identified genomic positions where the parents are homozygous for the same reference allele and the offspring heterozygous, a Mendelian violation consistent with a DNM. We filtered these candidate DNMs using current best practices, notably by checking that the non-reference allele is not present on the somatic sequencing reads in either parent (see Methods). In the end, we identified 202 putative DNMs in the 40 probands (i.e., the offspring of each of the 40 trios) (Fig 1).

To determine the parental chromosome on which the DNMs occurred, we employed two different strategies. For the 88 DNMs identified in the 18 probands with sequenced partners and offspring, we phased 91% of mutations based on their pattern of inheritance to the next generation (see Methods). For the 114 DNMs identified in the 22 probands without sequenced offspring, we phased by read tracing, i.e., by linking DNMs to phase informative heterozygous alleles found in the same sequenced DNA fragment (see Methods). Given the high genetic diversity in zebra finches ($\pi \approx 1\%$), we were

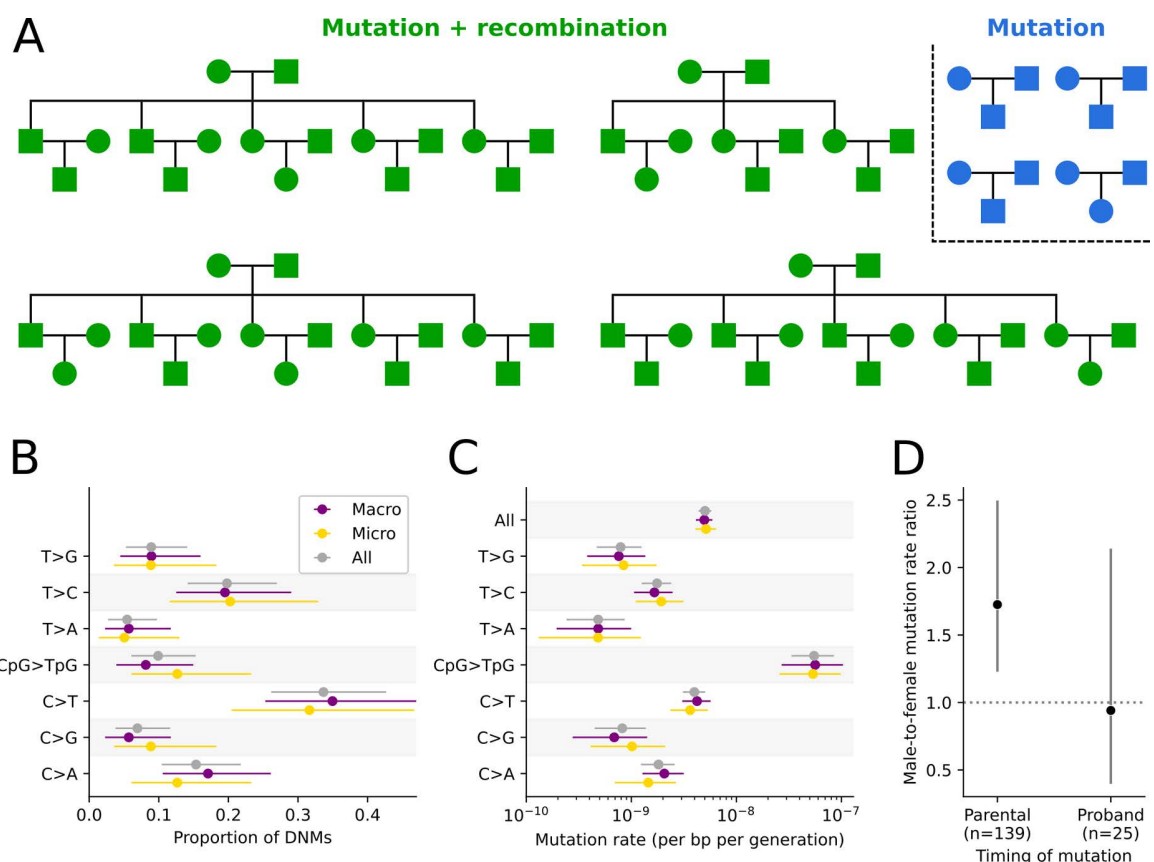

**Fig 1. Pedigree structure and mutation rate estimates in zebra finch.** (A) Familial relationships among the 74 sequenced zebra finch individuals. We identified DNMs in all families (blue and green), and studied recombination events in the nuclear families with multiple siblings (green) (see Methods). (B) Proportion of DNMs for the seven mutation types, in all autosomal sequences (gray), macro-chromosomes (purple), and micro-chromosomes (gold). (C) Mutation rate estimates for the seven mutation types and all mutations (top). Horizontal lines show the 95% CIs, assuming mutation counts are Poisson-distributed. (D) The ratio of male:female mutations for events occurring in the parental germlines or in early development of the proband. Horizontal lines show the binomial 95% CIs.

able to phase 81% of the 114 DNMs by this strategy. Applying both approaches to probands with sequenced offspring, 94.3% of the DNMs are assigned to the same parental chromosome as inferred by transmission, confirming the reliability of the phasing.

**Inferences of developmental timing of germline mutations.** Mutations that occurred in the early development of the probands can be mistaken for mutations that occurred in the parental germlines. To distinguish among these two possibilities, we looked for DNMs that showed "incomplete linkage" with neighboring heterozygous alleles, i.e., regions in which the (diploid) proband carries three distinct haplotypes (see Methods and S1 Fig for a visual example) [60]. Among the 174 DNMs with informative variants nearby (~150 bps, the length of our sequencing reads), 25 showed evidence of being post-zygotic, i.e., as having arisen after fertilization of the proband (see Methods). The fraction of post-zygotic mutations in zebra finches (~15%) is larger than that reported in humans (~5%) ($p < 3 \times 10^{-5}$ by a one-sided binomial test), consistent with previous studies showing larger proportions in shorter lived organisms like mice [16] and cattle [60]. In addition, these post-zygotic mutations are at lower than 50% frequency among the sequencing reads in the proband ($p < 10^{-7}$ by one-tailed Wilcoxon signed-rank test; S2 Fig), and as expected for mosaic mutations, have a transmission rate to the next generation significantly lower than 50% (21%, 95% CI: 7% - 47%). In contrast, the transmission rate for the rest of DNMs is 45% (95% CI: 33% - 55%), not significantly different from the 50% expected for a constitutive mutation (given complete detection power). Moreover, the 25 putative post-zygotic mutations occur at a similar rate on both parental chromosomes: the ratio of paternally to maternally phased mutations is 1.14 (95% CI: 0.33 - 2.34), again as expected if they occurred after fertilization of the proband. Collectively, these lines of evidence—under-transmission, lower allele frequency compared to constitutive heterozygous variants, and equal occurrence in both parental chromosomes—indicate this set of mutations indeed arose post-zygotically and are mosaic in the proband.

If a DNM occurs during or shortly after parental primordial germ cell specification, it may not be present in parental somatic cells but could be carried by a significant proportion of their gametes, increasing the likelihood of inheritance by multiple descendants. By analyzing the four multi-sibling pedigrees, we found only one instance of DNMs shared between siblings (S2 Table). While there are only a small number of siblings, finding only one such DNM suggests that the mutation rate during this early developmental stage is relatively low in zebra finches.

**Estimation of mutation rates.** Given the total length of genome sequence that we analyzed across the 40 trios (~40 Gbs) and the number of DNMs identified (202) (see Methods), the point mutation rate is $5.0 \times 10^{-9}$ (95% CI: $4.3 \times 10^{-9}$ - $5.7 \times 10^{-9}$; Fig 1C) per bp per generation. Dividing by the average parental age for both sexes in our pedigrees (2.5 years; S1 Table) yields an estimated mutation rate per year of $2.0 \times 10^{-9}$ per bp. The rate per generation is similar to what was previously inferred by pedigree sequencing in zebra finch: $5.8 \times 10^{-9}$, based on the average rates in two trios [32] and $6.2 \times 10^{-9}$, based on 16 trios [33]. It is also similar to what has been estimated for the collared flycatcher ($4.4 \times 10^{-9}$, based on seven trios; [30]), great reed warbler ($7.1 \times 10^{-9}$, based on eight trios; [31]), and 14 trios from four passerine species (ranging from $5.1 \times 10^{-9}$ to $6.9 \times 10^{-9}$; [32]). Thus, among these passerines with similar generation times (1–4 years), the per generation mutation rate appears to be quite stable.

An exact estimate of the mutation rate per generation would be based on a comparison of zygotes to germ cells. Instead, existing estimates, including ours, are based on somatic tissue samples from parents and offspring. This approach leads to the incorrect inclusion of DNMs that arose during early development of the probands [61]. In that respect, we note that if we only consider the DNMs inferred to have occurred in the parental germlines, our estimate reduces to $4.4 \times 10^{-9}$ per bp per generation. This rate, in turn, may be a slight underestimate of the mutation rate from zygote to germ cell, given that it does not account for early parental developmental mutations that are present at detectable allele frequencies in the parental somatic tissues and thus were excluded by our filtering steps [61] (see Methods).

The mutation rate is known to be influenced by the immediate sequence context, notably when a CpG site is methylated. To determine if DNMs in zebra finch reflect these and other influences, we classified all DNMs into seven single mutation types (Fig 1B and 1C). We found a transition:transversion ratio of 1.73 (a ratio not significantly lower than the value in

humans of ~2 [62], $p = 0.33$ by a two-sided binomial test), and observed that the CpG > TpG mutation rate is more than an order of magnitude higher than the rest of substitution types (Fig 1C), consistent with findings in vertebrates and beyond [63]. In addition, the proportions across the seven mutation types are highly similar to those observed in 19 million low-frequency polymorphisms (at most three copies, see Methods) segregating in a sample of 27 "unrelated" zebra finches (S3 Fig; $p = 0.11$ by a Chi-square goodness-of-fit test with 6 degrees of freedom). Since rare (young) variants are expected to reflect the mutational process, this close correspondence lends further support for the reliability of the DNM calls.

To explore if the mutation processes active in the mammalian germline also play a role in zebra finches, we performed an analysis using COSMIC single-base mutational signatures (SBS) [64]. In this approach, a mutation type is defined by the reference and mutated allele and the identity of the two flanking base pairs, and the mutational spectrum of a cell type or tissue is summarized by the proportion of the 96 mutation types. The mutation spectrum is then modeled as a linear combination of signatures, with the idea that each signature corresponds to one or few mutagenic processes. The COSMIC signatures were originally inferred from patterns of genetic variation in human tumor samples [65,66], and several were linked to known mutational mechanisms, such as exposure to environmental mutagens (e.g., tobacco smoke) or endogenous processes (e.g., CpG deamination) [66].

In the mammalian germline, two so-called "clock-like" mutational signatures, SBS1 and SBS5, have been shown to account for most mutations [18,34,35]. Specifically, most mutations are assigned to SBS5, of unknown etiology, and a smaller proportion to SBS1, which is thought to occur due to spontaneous deamination of methylated CpG sites [66]. In birds, both of these signatures are also present. In the 19 million relatively low frequency alleles, we found contributions from SBS5 (84% of mutations), SBS84 (14%), and SBS1 (2%) (S3 Table). When analyzing the smaller sample of DNMs instead (202 mutations), we observed contributions from SBS5 (59%), SBS19 (19%), SBS4 (15%), and SBS1 (6%) (S3 Table). These results suggest that SBS5, to a lesser extent SBS1, and possibly other mutation processes, play a role in the zebra finch germline, as reported for SBS5 based on polymorphism data in other taxa [67].

In contrast to most mammalian karyotypes, the genome of zebra finches and other birds contains a large number of micro-chromosomes (here defined as autosomes shorter than 40 Mb), which differ in GC content, replication timing, and other genomic features that could affect mutation rate [39,40]. Nonetheless, as far as we can tell, mutation rates (both the total and seven types) appear to be highly similar in both types of autosomes (Fig 1B and 1C).

Next, we compared the mutation rate between the sexes. In our pedigrees, parental ages are similar in both (mothers are only ~1 month older on average, S1 Table). For the subset of DNMs that occurred in the parental germline after primordial germ cell specification, we inferred a male:female mutation rate ratio of 1.72 (95%CI: 1.22 - 2.43) (Fig 1D). This estimate of the paternal bias in mutation is in good agreement with previous reports in birds based on putatively neutral substitution rates in sex chromosomes versus autosomes [68], but lower than that inferred from whole-genome sequencing of 16 zebra finch trios (~4; [33]). This difference could be at least partially explained by the younger average parental age at conception in our study (~2 compared to ~3 years in theirs), e.g., if there is a relatively fixed number of germline mutations per generation arising during parental embryogenesis that are not sex-biased [68].

To investigate whether a developmental window exists during which mutation rates are not sex-biased in zebra finches, we compared mutation rates between the sexes for the post-zygotic DNMs (using the sex of the proband instead of the sex of the parental haplotype; see Methods). After accounting for the different number of individuals for each sex among probands, the male:female ratio is 0.94 (95% CI: 0.43 - 2.02) (Fig 1D), consistent with there being no sex differences in the number of post-zygotic mutations, as also reported in humans [34], mice [16], and cattle [60]. Considering all mutations jointly, regardless of when they arose in development, the estimate of the male:female mutation rate ratio is 1.6 (95% CI: 1.19 -2.27).

## Meiotic recombination

**Detection of crossover and non-crossover events.** To detect autosomal crossover and non-crossover events, we used an approach based on the patterns of inheritance of informative sites along the multi-sibling, three-generation

pedigrees (Fig 1A). In brief, we relied on the configuration of informative sites (i.e., sites heterozygous in one parent but not in the other) to track changes of phase in parental haplotypes (following [69]). After filtering out regions in which genotype calls are unreliable (see Methods), crossovers were detected by an odd number of changes of phase within a short genomic distance; events involving more than one change of phase were classified as "complex" (see Methods). In turn, non-crossovers were detected by two changes of phase within a short genomic distance (see Methods).

In the 54 zebra finch meioses (28 maternal and 26 paternal) in which recombination events can be called (Fig 1A), we identified 1,088 autosomal crossovers. The high density of informative markers (~0.5%) allows us to delimit recombination events to a median interval of 606 bps (mean = 55 kb) (S4 Fig). Of these events, 559 and 529 occurred in maternal and paternal meioses, respectively, an average of 20.0 and 20.3 crossovers per meiosis. This difference is not statistically significant ($p = 0.76$ by a two-tailed binomial test) (S4 Table), in agreement with previous studies reporting an absence of heterochiasmy in zebra finch [70,71], and great reed warbler [56]. The absence of sex differences in these species stands in contrast with observations in other birds: e.g., recombination rates are higher in males in the collared flycatcher [55], the hihi [38], and the helmeted honeyeater [72], but higher in females in house sparrows [37]. Combining our data from both sexes in zebra finch, we find ~10% of crossovers are complex events involving more than a single change of phase, which is substantially higher than reports of ~1% in humans [7,73] and mice [6]. While this discrepancy might reflect distinct mechanisms of crossover resolution among taxa, it is also likely influenced by the analysis pipeline (e.g., HMM-based detection [6] vs. our approach (see Methods)) and in particular the criteria used to classify events as "complex" (e.g., phase changes within 100 kb [73] or within 10 informative sites (see Methods)). There is no discernable difference in the rate of complex events between sexes in our data ($p = 0.65$ by a Chi-square test).

Considering the number of crossovers per chromosome per meiosis, the estimated crossover rate is consistent with an obligatory crossover per tetrad [74,75] for 28 of the 39 chromosomes ($p < 0.05$; S5A Fig). Given that, in the absence of a back up mechanism for achiasmatic tetrads, we would only expect a couple of chromosomes to fall below this $p$-value by chance, this observation suggests that we are missing a few events, particularly on short and highly repetitive chromosomes (S5B and S5C Fig). Indeed, a study of MLH1 foci in zebra finch reported an average of 46 autosomal foci, which would correspond to 23 transmitted crossovers [76].

This caveat notwithstanding, the sex-averaged genetic map length is estimated to be 2,015 cM (1,996 cM and 2,035 cM for females and males, respectively), corresponding to a mean recombination rate of 2.11 cM/Mb. Despite the fact that this map length is likely to be a slight under-estimate, it is substantially higher than the two previous reports for zebra finch based on pedigrees: 1,068 cM (1.06 cM/MB) [70] and 1,341 cM (1.50 cM/Mb) [71]). Part of the explanation likely lies in which chromosomes were included, as most of the micro-chromosomes, which experience high recombination rates, were not included in earlier studies. Indeed, if we restrict our estimate to the same set of autosomal chromosomes as [71], our estimate (1.68 cM/Mb) is in better agreement with theirs. Moreover, previous estimates were based on few genetic markers (~900 and ~2,000, respectively), so may have missed a number of crossover events, even on larger chromosomes. The average rate for the six macro-chromosomes is 0.93 cM/MB–similar to estimates in placental mammals, which only have macro-chromosomes [77]. By contrast, the average rate for the 33 micro-chromosomes is 3.96 cM/Mb.

In parallel, we inferred a genetic map from patterns of linkage disequilibrium (LD) by combining data from six of the founders of the pedigrees and sequencing data that were previously generated for 19 unrelated zebra finches [51] (see Methods). LD-based genetic maps provide estimates of the population recombination $\rho = 4N_e r$, where $N_e$ is the effective population size and $r$ is the recombination rate per generation. Estimates for micro-chromosomes are likely less reliable, given the high background recombination rate (S6 Fig) [51]. For macro-chromosomes alone, the mean $\rho$ in our LD-based map is estimated to be 0.082 per bp. Given the observed diversity level in our sample (Watterson estimator $\theta_w = 0.0159$) and our estimate of the mutation rate, the estimated effective population size of zebra finch is 792,000. If we use this $N_e$ value to estimate $r$ from the population recombination rate, then the mean recombination rate is 2.8 cM/Mb on macro-chromosomes (S6 Fig). This estimate is of the same order as what we obtain directly from pedigrees but two- to

three-fold higher; such a discrepancy is not surprising given the numerous assumptions that come into play in estimating *r* from population data.

Next, we inferred non-crossover events by considering two phase changes among the parental haplotypes (see Methods). Given the mean tract lengths estimated in other vertebrates and the diversity levels in zebra finches, we expected such events to typically involve only a single informative variant, and thus for their identification to be highly sensitive to sequencing and genotyping errors [6–8]. In order to minimize the number of spurious non-crossover calls, we excluded the nine shortest micro-chromosomes, which together account for 1.6% of the assembled autosomal genome, and to which reads may be less reliably mapped (S7 Fig). We focused on the second generation of three generation pedigrees, because there are several siblings, allowing us to detect changes of phase. The F1 individuals from these crosses all have sequenced offspring, in which we can verify transmission of any putative non-crossover event. This subset of the data represents a total of 36 meioses (18 paternal and 18 maternal).

To make non-crossover calls, we tried three variant callers (*GATK*, *freebayes* and *bcftools*) and stringently filtered the results (see Methods). To estimate the reliability of the non-crossover calls, we quantified their transmission rate to the next generation (S5 Table), with the expectation that true events should be transmitted ~50% of the time (assuming complete power to detect transmission). On that basis, we determined that *freebayes* provides the most reliable set of non-crossover calls, with an observed transmission rate of 0.47 (95% CI: 0.38 - 0.57). Using this caller, we identified a total of 275 non-crossovers in the 36 meioses. As expected, most (235) of the non-crossover events involve a single informative site in the conversion tract (S8 Fig). As is the case for crossovers, there is no detectable sex difference: 141 were maternal and 134 paternal ($p = 0.72$ by a two-tailed binomial test) (S4 Table).

**Distribution of recombination events along the genome.** In many vertebrates, the crossover rates increase near telomeric regions, and this elevation is typically higher in males than in females [78–80]. The distribution of non-crossovers, however, remains poorly described, especially in non-mammalian vertebrates and species with micro-chromosomes. Considering the distribution of crossover resolutions in macro-chromosomes and micro-chromosomes separately, we found that rates are differentially distributed (Kolmogorov-Smirnov test $p < 2.2 \times 10^{-16}$; Fig 2). Whereas in macro-chromosomes, crossover rates are elevated towards the telomeres (as found, for example, in the guinea fowl [80]), in micro-chromosomes, they are much more uniformly distributed (Fig 2 and S6 Table for all *p*-values). Similarly, LD-based recombination rates, which primarily reflect crossovers, show an elevation near telomeres on macro-chromosomes, but not on micro-chromosomes (S9 Fig). A caveat, however, is that inferring recombination rates through LD-based approaches is challenging when background recombination rates are high [51,81], as may be the case for micro-chromosomes. In principle, the difference between macro- and micro-chromosomes could arise from a proportionally shorter effect of telomeres on large chromosomes.

One factor that may influence the placement of crossovers is the presence of polymorphic inversions. Estrildid finches are known to harbor many such inversions [82], including three large autosomal inversions in zebra finch [83]. As expected, individuals that we infer to be heterokaryotic have no crossovers within the putative inverted regions, when the same is not true for inversion homozygotes (S10 Fig).

In 98 instances, more than one crossover occurred on a chromosome in a given meiosis. Comparing the distribution of distances between crossovers occurring within the same meiosis ("within") to what is seen for events occurring in different meioses ("between"), the "within" distribution is shifted towards longer distances (S11 Fig), as expected from crossover interference. Interestingly, however, the "within" distribution appears to be bimodal, with an enrichment of crossover events separated by shorter distances (~$10^5$ bps). Specifically, in 32 cases (~3% of the total 1,088 crossovers), one event is located within 1 Mb of another (i.e., the minimal distance between the outer edges of the two crossover intervals is < 1 Mb). Given that these crossovers are well supported and stringently filtered, they appear to be genuine. One possibility is that they represent class II crossovers, which are not subject to crossover-interference and have been reported at similar but slightly higher proportions in other species (e.g., 5–10% in mice [84] and 5% in *Arabidopsis thaliana* [85]).

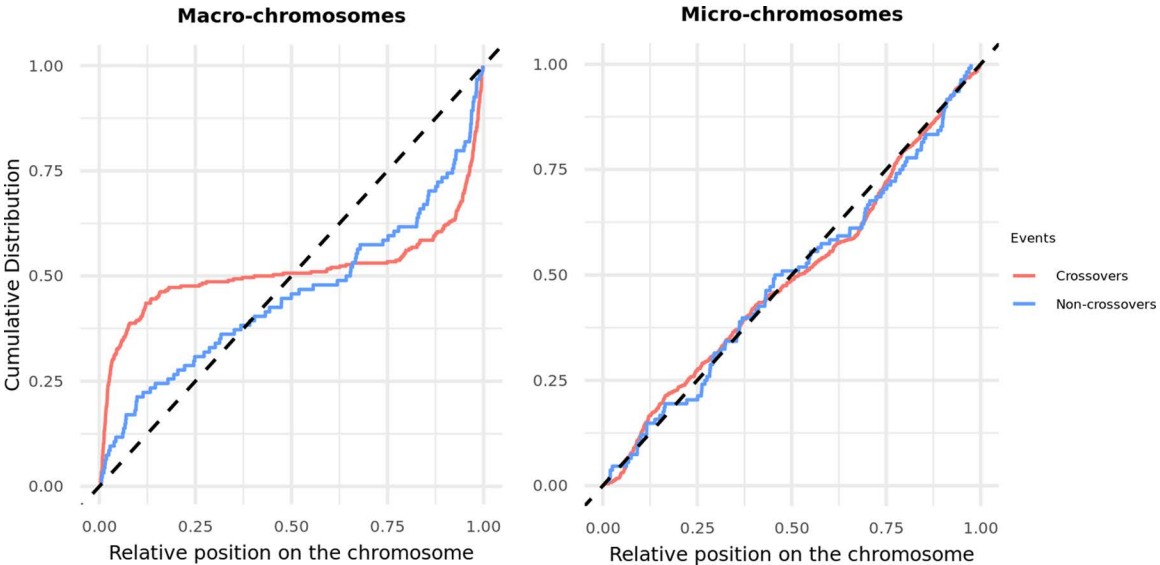

**Fig 2. Cumulative distribution of recombination events along both types of chromosomes.** The cumulative distribution of crossovers (in red) and non-crossovers (in blue) on the autosomes, for macro-chromosomes (left) and micro-chromosomes (right). Macro-chromosomes are defined as autosomes longer than 40 Mb. The position of the events is normalized by the length of the chromosome (see Method). The black dashed lines represent the uniform distribution. S13 Fig shows similar results, separating micro-chromosomes above and below 20 Mb in length. S6 Table reports p-values from Kolmogorov-Smirnov tests for various comparisons (crossover versus non-crossover, macro versus micro-chromosomes).

However, we cannot exclude the possibility that a subset of the apparent double crossovers in close proximity instead result from long gene conversion events (as reported at frequencies of ~2% in primates by [8,13,86]), or complex resolutions with template switching (as found at low frequencies in primates by [7,8,86]).

In contrast to crossovers, the spatial distribution of non-crossovers does not differ between micro- and macro- chromosomes ($p = 0.16$, Fig 2 and S6 Table). Accordingly, the distribution of crossovers along the chromosomes is significantly different from that of non-crossovers on macro-chromosomes ($p = 4.8 \times 10^{-6}$), with less of a skew towards telomeres among non-crossovers, while the same is not seen in micro-chromosomes ($p = 0.36$) (Fig 2 and S6 Table). None of these patterns differ significantly between sexes (S12 Fig).

**Overlap with genomic features.** Zebra finch, like other birds, lacks a functional copy of PRDM9 [41,87]. Consistent with findings in mammals without PRDM9, meiotic recombination is enriched in promoter-like features [51,52,88]. Specifically, analyses of LD suggest that the increase in recombination rates appears to be mainly attributable to CpG islands, with no further increase explained by TSSs [51]. To test this hypothesis more directly, we first improved the annotation of CpG islands in zebra finch. To this end, we relied on the fact that CpG islands are usually hypomethylated in vertebrates [89] and quantified DNA methylation levels in the testes of two males using bisulfite sequencing (see Methods). We then considered both DNA methylation levels and local sequence composition in order to identify 46,205 CpG islands, on par with numbers reported for other vertebrate species [46,90].

Using this annotation, we asked whether the recombination events identified in pedigrees are enriched at CpG islands. As expected, significantly more recombination events occur close to a CpG island than expected by chance: 23.7% of crossovers occur within 100 bps of a CpG island when only 17.5% are expected to do so by chance on average (a 1.36-fold enrichment), and 17.8% of non-crossovers when 11.5% are expected to do so by chance (a 1.54-fold enrichment) (Fig 3). Given that the methylation data is from testes, we redid this analysis restricting ourselves to only recombination events in males. The qualitative result is the same: the observed overlap remains significantly higher, albeit slightly lower, than expected by chance (see S14 Fig). The fact that we expect (and observe) more overlap between crossovers and

CpG islands than we do for non-crossovers likely reflects a difference in the genomic distribution of the two types of recombination events: crossovers are relatively more likely to occur in telomeric regions and micro-chromosomes (Fig 2), which have increased CpG island densities (S15 Fig) [91]. A rough calculation suggests that the degree of overlap is consistent with all crossover hotspots occurring at CpG islands: if each CpG island is assigned the mean heat inferred for hotspots in our LD-based map (9.93-fold), then given that CpG islands (+/- 100 bps) cover 3.48% of the autosomes, we would expect 26.4% of crossovers to overlap CpG islands, when we observe 23.7%.

As previously reported on the basis of LD patterns [51], recombination events are not enriched at TSSs conditional on the presence of a CpG island nearby (whereas they are enriched at CpG islands irrespective of the presence of a TSS) (S16 and S17 Figs). To verify the generality of this finding, we conducted an analogous analysis using the 443 recombination events called in collared flycatchers [55] and identified CpG islands with *cpgplot* (default parameters, [92]). As shown in S18 Fig, the enrichment of recombination events at CpG islands, and not at TSSs, is very similar to what we see in zebra finch. Therefore, in birds as in rodents lacking PRDM9 [48,50], proximity to a CpG island is predictive of both crossover and non-crossover events.

Large-scale analyses in humans have reported that meiotic recombination is mutagenic, with 1/200 mutations arising from a DSB [73,93]. Given the relatively small number of *de novo* mutations identified in our study, we should have very limited power to detect an effect in birds of a similar magnitude. Accordingly, we did not find evidence for the co-occurrence of *de novo* mutations and crossover events in zebra finch (S19 Fig; see also [33]).

**Estimation of fundamental parameters of gene conversion.** To the best of our knowledge, GC-biased gene conversion (hereafter gBGC) has only been examined directly in one bird species, the collared flycatcher [55], for which the authors estimated that 59% of events that include a AT/GC polymorphism are resolved towards GC rather than AT (CI: [52–65]). This point estimate is on par with what has been reported for mice and primates (57%-68%; [6,7,9,11,12,86]).

A challenge in estimating gBGC is that even a small fraction of false positive calls can bias the estimate downwards (i.e., towards 50%). To minimize the problem, we only used the events detected in the three pedigrees with five siblings, which appear to be the most reliable (S7 Tables). This approach led to the identification of 191 non-crossovers. We also considered the subset of non-crossover events that occur less than 100 bps from a CpG island, reasoning that such events are more likely to be true positives [51] (Fig 3).

Considering the 177 cases of AT/GC variants within all the non-crossover events, the gBGC bias in zebra finch is estimated to be 59% (95% CIs: 52–66), the same point estimate as reported in flycatchers [55]. Moreover, we can reject a null model of no GC-biased gene conversion for zebra finch (p = 0.008, by a one-tailed binomial test). When focusing on the 18 cases when the non-crossover event is close to a CpG island, the point estimate is higher but with large uncertainty (72%; 95% CIs: 47–90).

A recent study reported that the gBGC depends on the number of heterozygous sites present in the conversion tract: specifically, a bias was only seen for conversion events with a single such site both in mice and, more tentatively, in humans [6]. Two recent studies, one focusing on humans and the other on humans, chimpanzee and gorilla found no evidence for this claim [11,12]. To examine this question in zebra finch, we separated the non-crossovers into those with one heterozygous site versus more than one (see Methods). We also find no evidence supporting the hypothesis that gBGC depends on the number of heterozygous sites in the conversion tract, but the data are insufficient to reach a firm conclusion (Fig 4A).

Next, we estimated the mean conversion tract length, following [6]. Specifically, we relied on the distances between co-converted and non co-converted informative sites and assumed a single exponential distribution of tract lengths (see Methods). While the assumption of an exponential distribution is not valid, previous results in mammals suggest that it is a reasonable approximation for the vast majority of non-crossover events and it allows us to compare our findings to those previously reported [6,8,12,13]. In baboons and humans, it appears that a tiny fraction of events (<2%) are kilobases in length and likely arise from a distinct process [8,12]. While the small number of non-crossover events prevents us from

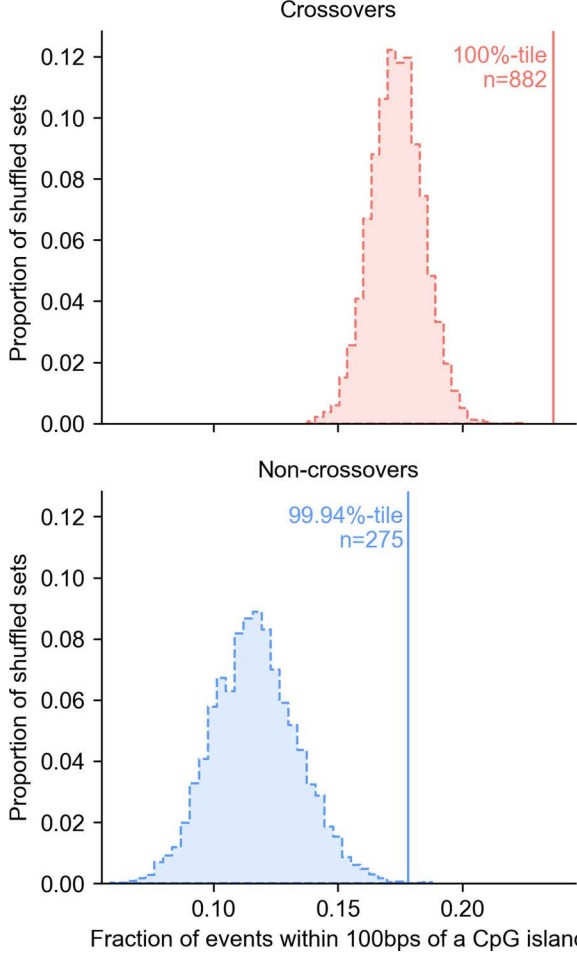

**Fig 3. Overlap of both types of recombination events with CpG islands.** Fractions of crossovers (top) and non-crossovers (bottom) within 100 bps of a CpG island. The vertical lines show the observed overlap. The distribution for the overlap expected by chance is shown as a histogram, obtained by randomly shuffling all the events within a 2.5 Mb window on each side of their original location, matching for the GC content and ensuring similar mappability (see Methods for details).

exploring that possibility in depth, there are a couple of cases where the minimum tract length is likely kilobases in length: for example, among the tracts that include more than one informative site, the distance between sites is 642 bp in one case, and 4,169 bp in another. (A subset of the events that we classify as crossovers occurring in close proximity, which we hypothesize to be class II events, may also be long non-crossovers; however, they would have to be very long, i.e., on the order of ~100 kb (S11 Fig).) If we exclude the event that is minimally 4 kb in length, the mean conversion tract length of non-crossovers in zebra finch is estimated to be 32 bps on average (central 95%-tile: 24–42 bps; see Methods). These results are comparable to a number of previous estimates in mammals [6,8,11–13].

**Total number of non-crossovers per meiosis and non-crossover to crossover ratio.** Given the mean conversion tract length and the density of informative sites, we can infer the expected number of non-crossovers per meiosis. This inference relies on our power to detect a non-crossover, i.e., on the probability that a gene conversion event overlaps an informative site, which in turn depends on the mean conversion tract length and the density of informative sites (see Methods). We inferred 191 non-crossovers in 30 meiosis or 6.4 events per meiosis (considering only families of five siblings; S7 Table). For a mean gene conversion tract length of 32 bps, we should detect an estimated 6.9% of

non-crossover events in the genome, averaged over all founders. In other words, the total number of non-crossover events per genome (*i.e.*, chromatid) is estimated to be~14.9-fold higher than observed. Taking into account differences in power among founders, our prediction is that there are 92.4 non-crossover events per genome per meiosis on average; given the uncertainty in the conversion tract length estimate, between 73.4 and 117.3 non-crossover events (Table 1).

In turn, our estimates of the number of non-crossovers suggest that the non-crossover:crossover ratio per genome is 5.4 (between 4.3 and 6.9, given the tract length uncertainty). This estimate of 5.4 (or 10.8 if considering the ratio per tetrad) is similar to the one in mice [94].

The densities of crossovers and non-crossovers both decrease with chromosome length (Fig 5A). If we ignore the subset of events repaired off of the sister chromatid, which leave no discernable mark, or through mechanisms other than homologous recombination [75], and consider the number of DSBs to be given by the sum of crossovers and non-crossovers, it follows that the same is true of DSBs (Fig 5B). Thus, either larger chromosomes have a lower density of DSBs or a larger fraction of DSBs are resolved non-canonically (i.e., as neither a crossover nor a non-crossover). While the density of DSBs appears to decrease with chromosome length, the estimated number of DSBs increases (Fig 5C).

## Discussion

To date, meiotic recombination events and DNMs have only been directly identified in a handful of non-mammalian verte-brate species, based on a small number of trios. There is therefore little comparative information about properties of mutation and recombination, beyond average rates in the genome per generation. In this study, we collected whole genome sequences from 74 zebra finches. The genomic data provided information about 80 meioses, and allowed us to infer key mutation and recombination parameters.

As we had previously reported based on comparisons of substitution rates on the Z chromosome and autosomes for passerines, we found that the paternal bias in germline mutation is lower in zebra finches (1.6:1, S2 Table)

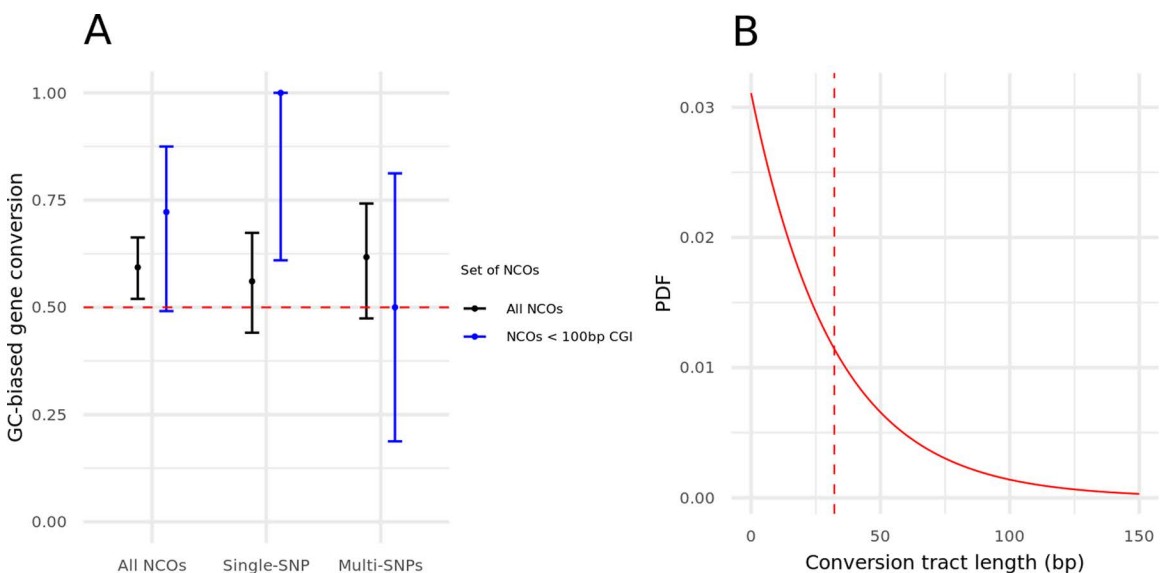

**Fig 4. GC-biased gene conversion and conversion tract length distribution estimates.** (A) Point estimates of the GC-biased gene conversion (gBGC) for different sets of non-crossovers. The 95% CI were obtained from an exact binomial test (see Methods). The black dots are the point estimates for the set of all non-crossovers and the blue dots are the estimates for non-crossovers within 100 bps of a CpG island. For each set of non-crossovers, we looked either at (i) all the events, (ii) the event with only one heterozygous site in the conversion tract, (iii) the events with more than one heterozygous site. (B) Estimated distribution of the conversion tract lengths. The mean is 32 bps, indicated with a vertical red dashed line.

than in most mammals (about 3:1, [68]). This difference could arise from lower rates of endogenous DNA damage in avian males or from a greater proportion of DNMs occurring during early developmental stages, when both sexes have similar mutation rates (Fig 1D). Our estimate is lower than the ratio of ~7.5:1 (95% CI: ~4.5:1 - ~13.5:1) reported for passerines in a recent study, which pooled phased germline mutations across four different species (including zebra finch) [32], as well as zebra finch-specific estimate of ~4:1 [33]. However, other studies in passerines have reported lower paternal biases, in better agreement with our estimate: ~1.5:1 in collared flycatcher [30] and ~3:1 in the great reed warbler [31]. In principle, variation among estimates could reflect, at least in part, differences in the parental ages of the sequenced pedigrees. Regardless, sex-averaged per year and per generation mutation rates in zebra finch and other passerines are similar to what is seen in mammals with the same generation time [32] (S20 Fig).

These data also allowed us to characterize crossovers and non-crossover events. As expected, the constraint of an obligate crossover per chromosome leads to higher rates of recombination on micro-chromosomes. On macro-chromosomes, however, our estimate of the mean recombination rate for the zebra finch is substantially lower, and comparable to that of mammals (S6 Fig) [77]. At a finer-scale, as seen in dogs as well as in mouse and rat knockouts for PRDM9 [48]–[50], recombination events are enriched near CpG islands (Fig 3). Moreover, despite the absence of PRDM9, properties of recombination resolutions—including the mean gene conversion tract length, the degree of GC-biased gene conversion, and the ratio of crossovers to non-crossovers—are all comparable to what has been reported in mice and humans (Fig 4 and Table 1).

Thus, despite different mechanisms guiding recombination to the genome, fine-scale recombination parameters in zebra finch are similar to those in mammals, as are mutation rates per generation and mutational signatures in the germline. Mechanistic constraints alone seem unlikely to account for this conservation, given that mutation and recombination rates are heritable [20,86,95–97], alleles affecting these processes are known to segregate in mammals [20–22,73,98,99], and some of the parameter values estimated here differ markedly in yeast species, *A. thaliana*, or *Drosophila* species, for example [2–4,100,101]. The similarity in rates over vast phylogenetic differences instead points to a role for stabilizing selection, with similar optimal recombination and mutation parameters across these vertebrate lineages.

The increasing availability of genome sequences should make it possible to test this hypothesis more systematically, and to identify which aspects of mutagenesis and meiotic recombination are most strongly conserved. As this study demonstrates, many of these parameters can now be estimated in non-model species by sequencing a relatively modest number of large three-generation pedigrees. Recent advancements in long-read sequencing offer further possibilities: in particular, *high-fidelity* (HiFi) long-read sequencing of germ cells will enable the detection of a large number of mutation and recombination events [10–13,102]. Application of such approaches across the tree of life will help address long-standing questions about the selective pressures that shape the evolution of mutation and recombination [23,103–109].

Table 1. **Estimates of number of non-crossovers per chromatid. Predicted number of non-crossovers and estimated non-crossover to crossover ratio for different estimates of the mean conversion tract length. We report the ratio per chromatid (i.e., the number of predicted non-crossovers divided by the number of observed crossovers); we note that this ratio is sometimes reported per tetrad instead [94]. The number of crossovers reported in this table is based on the 30 chromosomes in which non-crossovers were identified (see Methods).**

| Tract length estimate | Observed number of crossovers | Predicted number of non-crossovers | NCO:CO per chromatid |
|---|---|---|---|
| 24 | 17.1 | 117.3 | 6.9 |
| 32 | 17.1 | 92.4 | 5.4 |
| 42 | 17.1 | 73.4 | 4.3 |

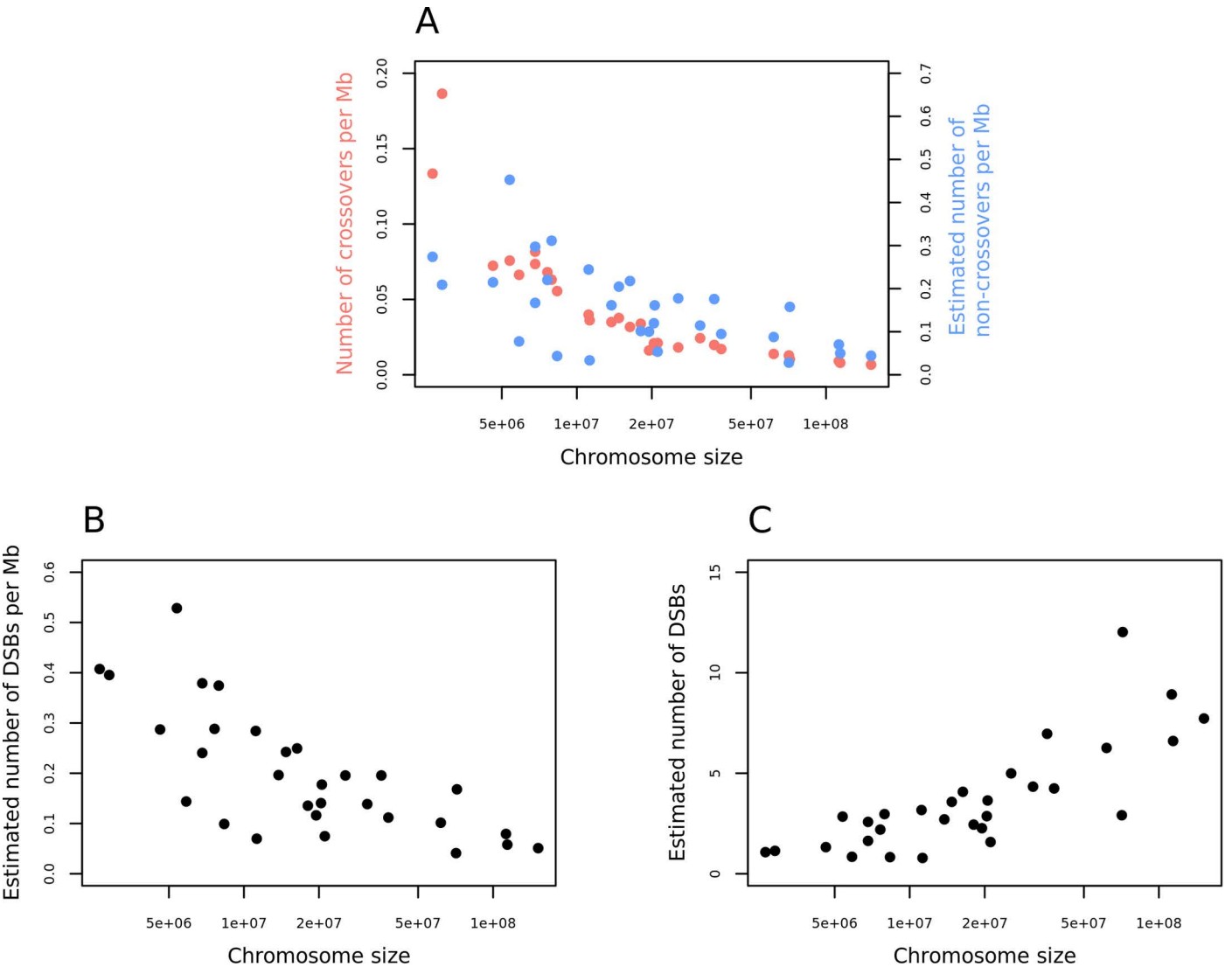

**Fig 5. Densities of recombination events per chromosome.** (A) Density of crossovers (blue) and non-crossovers (red) as a function of chromosome size for the 30 autosomes that passed mapping quality and coverage thresholds (see Methods). (B) The estimated density of DSBs as a function of the chromosome size, for the same set of chromosomes. (C) The estimated number of DSBs as a function of the chromosome size.

## Materials and methods

The command lines used for each of the following subsections can be found in the S1 Text.

### Ethics statement

The ancestors of the zebra finch pedigrees sequenced here were taken from the wild at Fowlers Gap (New South Wales in southeastern Australia), under NSW Scientific license SL100378. Wild-derived zebra finches were held and bred at Macquarie University (Sydney, Australia), under Animal Ethics Approval 2018/027.For the bisulfite sequencing, all procedures were conducted in accordance with the NIH Guidelines for the Care and Use of Animals and were approved by the University of Chicago Institutional Animal Care and Use Committee (ACUP #72220).

## Pedigree breeding and sequencing

Zebra finch pairs were bred in one of 20 outdoor aviaries (4.1 × 1.85×2.24 m) with a single pair per aviary, with access to *ad libitum* water, finch seed mix (Avigrain Finch Blue, Berkeley Vale, Australia), and a 'greens' mix (blended spinach and frozen vegetables, as well as micro- and macronutrient supplements from Naturally For Birds, Pullenvale, Australia). Each aviary had three nest boxes and nesting material ('November' grass and emu feathers). Blood was collected from F0, F1 and some F2 individuals via venipuncture of the brachial vein and stored in 99% ethanol. Tissue samples for the remaining F2 individuals were collected from terminated embryos between day 5–10 of incubation (euthanised via decapitation). Legs and wings of the embryos were stored in 99% ethanol before DNA extraction.

Whole blood (n = 53) and whole tissue (n = 21) samples were shipped to Genewiz (Azenta Life Sciences, South Plainfield, NJ, USA) for DNA extraction, library preparation, and whole genome sequencing. To reduce the possibility of false positives due to amplification errors, the 150 bp paired-end DNA libraries were prepared using a PCR-free protocol. The libraries were sequenced on an Illumina Hiseq X with a target of ~350 million reads per lane. Genotype calling errors in the parental generation can greatly increase the false positive rate during mutation calling, as true heterozygous sites that have been incorrectly genotyped as homozygous will appear to be a DNM. To guard against this possibility, we targeted the parental generation for higher coverage sequencing relative to the other individuals in the pedigree.

## Whole genome sequencing alignment

We combined the whole genome sequences that we generated from the pedigrees (S1 Table) with previously generated genome sequences for 19 "unrelated" individuals sampled from an Australian population located in the same area as the eight founders of the pedigrees [51]. After removing adapters from all sequencing reads using *cutadapt* [110], we mapped them to the zebra finch reference genome sequence bTaeGut1.4 (NCBI RefSeq ID: GCF_003957565.2) using the Nextflow pipeline *sarek* (v 2.7.1) with default parameters [58,111,112]. The familiar relationship among individuals, along with the mapping coverage for each sample, are presented in S1 Table. Given that the sex chromosomes have a lower depth of coverage in the heterogametic sex, making it harder to reliably detect DNMs and recombination events, we focused our analyses on autosomal sequences, as had been done in previous studies (e.g., [30,32,55,113]).

## Variant calling

We called variants using GATK (version 4.2.6.1; [114]). First, *HaplotypeCaller* was used to generate per sample GVCF files for each assembled chromosome [114], including non-variant sites and setting the heterozygosity parameter to 0.01 (version 4.2.6.1; [114]). We then combined GVCFs from each sample using *GenomicsDBImport* and called a final set of genotypes using *GenotypeVCFs*, setting again the heterozygosity to 0.01 (version 4.2.6.1; [114]). With this approach, we identified a total of 101 million single-nucleotide polymorphisms (SNPs) and 17 million indels.

## Mappability

To restrict our analyses to locations where short read sequences can be confidently mapped, we built a mappability mask for the zebra finch genome assembly following the *SNPable* pipeline, with a read length of 150 bps and stringency parameter of 1 (version 0.2.4; https://lh3lh3.users.sourceforge.net/snpable.shtml). We then used the script makeMappabilityMask.py from *msmc-tools* (Git commit 6b59e8e) to generate a final BED file of mappable genomic regions [115]. The mappability mask excludes 33.4% of the genome assembly. We removed all single nucleotide variants closer than 5 bps of an indel. In mappable regions, we called a total of 45.6 million SNPs of which we retained the 43.5 million that are biallelic.

## Detection of single-point *de novo* mutations

To detect DNMs, we analyzed the genomes of the 40 parents-offspring trios in the seven pedigrees (Fig 1A and S1 Table), looking for mutations in the proband and not found in their parents. For each trio, we examined which positions fulfilled all

the following requirements using *bedtools intersect* (version v2.30.0; [116]): (i) all individuals in the trio had a depth of coverage of at least eight reads and at most two times their autosomal average coverage, (ii) the position was at least 5 bps away from an indel called in any individual in the trio, (iii) it was not found within repetitive regions annotated in the reference assembly, and (iv) it was within the mappable section of the genome (see Mappability section). By this approach, we retained an average of 502.8 autosomal Mb per trio (min = 475.3, max = 511.1), which represents 52% of the autosomal genome.

For each trio, we then identified positions with a putative DNM as those that were heterozygous in the proband, homozygous for the reference allele in both the parents, and not polymorphic in all genotyped individuals outside that nuclear family. While DNMs could theoretically occur at sites that are polymorphic in the parents (e.g., due to recent back mutations), this effect is expected to minimally impact our estimates, as only 1.5% of all callable sites across trios have parents that are not homozygous for the reference allele (as expected from the ~1% heterozygosity; [59]). Furthermore, we only considered biallelic sites and required the genotype quality to be at least 30 in all individuals in the trio. A potential source of false positives is inherited variants that were erroneously called as homozygous in the parents. To minimize this problem, we discarded any candidate mutation if *GATK* reported the presence of a read-based assembled haplotype supporting the non-reference allele in either parent, regardless of their called genotype. We noticed that the local reassembly of reads performed by *GATK* to improve variant calling sometimes "erases" the presence of rare reads carrying the non-reference allele; we therefore verified the absence of reads carrying the DNM in the parents using *bcftools mpileup* (version 1.9; [117]).

Another potential source of error is when a homozygous reference allele is miscalled as heterozygous in the proband. To mitigate this issue, we required at least two reads carrying the non-reference allele in the proband, and discarded sets of putative DNMs within 10 bps of each other. While a tiny fraction of germline mutations in humans have been shown to occur in close proximity [118,119], this filter allows to discard false positives resulting from alignment to collapsed paralogous regions in the reference assembly [120].

### Assignment of parent of origin of *de novo* mutations

To determine the parent of origin for each DNM detected in the 18 probands with sequenced partners and descendants, we phased alleles by transmission. Specifically, we searched for informative sites within 200 kb of each DNM. An informative site was defined as position where an allele is carried by only one of the two parents (either as in a heterozygous or homozygous state); present in the proband (who is therefore heterozygous); not carried by the partner; and carried by the descendant. We then assigned a parental origin to the DNM when at least 75% of all informative alleles came from a single parent: if the DNM was transmitted to the descendant, it was assigned to that parent, and if not, it was assigned to the other parent.

Additionally, we phased DNMs detected in the 22 probands without sequenced descendants using 'read tracing' [121,122]. Briefly, for each DNM, we first searched for informative sites nearby (± 300 bps), i.e., positions where an allele inherited by the proband is carried by only one of the parents. If the DNM was carried on the same read pair as the inherited allele, we assigned the DNM as having occurred in the germline of the parent carrying the informative allele. In turn, if the DNM was not on the same read as the informative allele, we inferred that the DNM occurred in the other parental germline.

### Distinguishing mutations by their developmental timing

Mutations that occurred in the early development of the offspring can be mistakenly identified as mutations that occurred in the cell lineage leading to the parental germlines. To distinguish these events, we looked for candidate DNMs that showed "incomplete linkage" with informative heterozygous alleles nearby. In particular, we identified all heterozygous sites within ± 300 bps from a DNM in an offspring. In 37 of the mutation events, there were no neighboring heterozygous positions within this distance. For the rest of mutations, we searched for indications of three distinct haplotypes in the offspring by assessing the linkage of the mutation and nearby heterozygous alleles (S21 Fig). By this approach, we

discovered that 25 of the apparent parental germline DNMs exhibited evidence of being post-zygotic mutations that arose after the fertilization of the proband (see main text).

We also detected 32 DNMs in which reads assembled into more than two haplotypes in the parents (S21 Fig), violating the expectation for diploid individuals, and suggesting genotyping errors. We discarded such events.

### Estimation of the mutation rate per generation

We calculated an autosomal point mutation rate per site per generation by dividing the number of DNMs by two times the length of the autosomal genome considered (see "Detection of single-point *de novo* mutations"). To calculate mutation rates for specific mutation types (e.g., C>T), we divided by two times the number of mutational opportunities of that type in the subset of the genome considered (e.g., C). The 95% confidence intervals for the mutation rates were calculated assuming a Poisson distribution, based on the number of observed mutations and the total sequence length.

We note that, while mappable regions are homogeneously distributed along the zebra finch genome (S26 Fig), our data can only estimate mutation rates in these regions. Recent pedigree studies in humans based on high-accuracy, long-read sequencing technologies reveal mutation rates to be elevated in low-complexity regions [10]. Whether this observation extends to avian species remains an open question, requiring further investigation with sequencing approaches that allow a more comprehensive analysis of the genome.

### Mutational spectrum in low frequency polymorphisms

To analyze mutation types in polymorphism data, we considered low-frequency SNPs in a sample of 27 non-closely related zebra finch individuals (i.e., polymorphic sites where the alternative allele is found at most in three chromosomes out of the 54 chromosomes of the 27 individuals), using the *--max-maf* parameter in *VCFtools* (version 0.1.16; [117]). We then classified these mutations in seven distinct substitution types (see Fig 1), assuming the ancestral allele is the major allele, and using the flanking nucleotides in the reference assembly to distinguish C>T mutations that occurred in CpG context.

### Inference of mutational signature activity

We inferred COSMIC mutational signatures [66] from low-frequency polymorphisms and DNMs identified by pedigree sequencing. To this end, we classified mutations into 96 substitution types depending on their trinucleotide context in the zebra finch reference assembly, and generated mutation matrices with the number of events for each type. To infer signature activity, we used the *cosmic_fit* function in *SigProfilerAssignment* (v0.0.30; [123]), which assumes the genome-wide frequencies of the 32 trinucleotide contexts are consistent with those observed in humans. To account for the differences between human and zebra finch genomes, we adjusted the mutation counts by scaling them according to the ratio of genome-wide trinucleotide frequencies. Specifically, for each mutation type, we multiplied the observed count by the ratio of the trinucleotide frequency in zebra finches to that in humans.

### Co-localization between recombination and mutation events

We calculated the distance between each phased DNM and the closest crossover or non-crossover recombination event, discarding five mutations that were phased both by read-tracing and transmission but were assigned to a different parent (see Methods). We took this approach for events that occurred in the germline of the same parent and were detected in the same proband ("within"), as well as for events that occurred in different parents and probands ("between"). For cases where no recombination event occurs on the same chromosome as a DNM, we set the distance to 200 Mb, i.e., longer than the largest chromosome in the zebra finch assembly. To determine whether mutation and recombination events co-localize more than expected by chance, we compared the distribution of "within" and "between" distances by a Kolmogorov-Smirnov test.

**Individuals used to estimate diversity levels and infer an LD-based genetic map**

Using the biallelic SNPs in mappable regions, we obtained the Weir and Cockerham's Fst estimator between the individuals sequenced by [51] and the founders in our pedigrees with *VCFtools* [117,124]. We calculated Fst in 10 kb windows; the average Fst was 0.009. Given the low levels of genetic differentiation, we combined the two sets of samples for analysis, for a total of 27 zebra finches.

We estimated the population mutation rate $\theta=4N_e\mu$, where $N_e$ is the effective population size and $\mu$ the mutation rate per generation, using Watterson's estimator $\theta_w$ for the 27 finches [125]. We used *bedtools coverage* to count the number of biallelic segregating sites in regions within autosomal mappable regions, then divided this count by the total length of these regions (v2.30.0; [116]). With this approach, we estimated $\theta_w=0.0159$; given a mutation rate estimate of $5.02 \times 10^{-9}$ (see main text), this yields an estimated $N_e$ of ~792,000.

Because LDhelmet can take as input a maximum of 50 haplotypes, we based the inference of LD-based recombination rates on a subset of 25 individuals. To remove two from the 27, we used the *relatedness2* option in VCFtools on the same set of biallelic SNPs as in the Fst analysis (see S22 Fig for distribution of pairwise relatedness). We chose the two (both founders of the pedigree) so as to decrease the highest pairwise relatedness between the individuals used in the LD-based recombination rate; after excluding them, the maximal value of the kinship coefficient decreased from 0.09 to 0.032.

**Ancestral alleles and the mutation transition matrix**

To infer the ancestral allele state at biallelic sites, we tested if the frequency of the major allele among the 27 non-closely related individuals significantly exceeded 0.5 (exact binomial test, one-sided). In cases where the major allele frequency was significantly greater than 0.5 (96.5% of the SNPs), we assigned to the major allele a probability of 0.91 of being the ancestral allele, and a probability of 0.03 to each of the three remaining alleles. For SNPs where the major allele frequency did not significantly exceed 0.5, we assigned a probability of 0.47 to the two observed alleles, and a probability of 0.03 to the two other alleles. These polarized SNPs were then used to compute the mutation transition matrix following the method described in Chan *et al*, (2012) [126].

**Variant phasing**

We phased variants using information from sequencing reads that span multiple heterozygous sites (called "phase informative variants", PIRs) and the patterns of haplotype inheritance across the pedigrees. The PIRs were extracted with the SHAPEIT suite and a file with the familial relationships in the pedigrees was formatted using plink2. Variants were then phased using the *assemble* tool in SHAPEIT, and the outputs were converted into the LDhelmet VCF format (v2.r904; [127]).

**Implementation of LDhelmet**

We used the estimate of $\theta_w$ and of the effective population size to run LDhelmet (version 1.9; [126]). Apart from the set of *p* values (*i.e.,* the population scale recombination rate; see command line section "LDhelmet analysis" of the S1 Text for *p* values), we used the recommended parameters for all the steps of LDhelmet. For the Markov chain Monte Carlo analysis, we used three block penalties (10, 20 and 50), a burn in of 100,000 steps and a total of one million steps.

To identify LD-hotspots, we computed the mean recombination rate in windows of 1 kb, and assessed the relative recombination rate of each window compared to the mean recombination rate of the 20 kb each side, with a buffer of 2 kb between the focal window and the surrounding region [46]. Windows with a relative recombination rate greater than five were kept, and merged if adjacent. We identified 5,241 hotspots, on par with previous results in finches [51]. These LD-based hotspots are highly enriched at CpG islands (S23 Fig), as expected from previous LD-based maps and from our findings for crossovers and non-crossovers (Fig 3) [49,51].

Previous simulations indicate that there is limited power to detect hotspots in zebra finch where the population recombination rate is high [51]. One implication is that we expect to detect many fewer hotspots in telomeric regions of macro-chromosomes, as well as on micro-chromosomes. Perhaps for that reason, there is only a weak overlap of crossovers and non-crossover events with LD-based hotspots (S24 Fig). As this observation makes clear, although LD-based inferences have been transformative to our understanding of recombination (e.g., [109]), it remains useful to complement them with the direct identification of recombination events.

### Identification of CpG islands

We quantified the methylation status of CpG sites with bisulfite sequencing. To this end, we used testis samples from two male zebra finches. Samples were kindly provided by Sarah London (University of Chicago). Birds were bred in the London laboratory flight aviaries, and housed on a 14:10 h light:dark cycle. Food and water were provided *ad libitum*. Testis samples were dissected and flash-frozen in liquid nitrogen. Sequencing was performed by Azenta Life Sciences (Indianapolis) on a Illumina NovaSeq 6000 (PE150 technology) to generate ~50 Gb per sample. We processed and analyzed the bisulfite sequencing with the Nextflow pipeline *methylseq* (v 11.0.13) with default parameters [128].

CpG islands are regions in the genome that contain a large number of hypomethylated CpG dinucleotide repeats and often serve as sites of transcription initiation [89]. We initially used the *cpgplot* [92] tool in EMBOSS:6.6.0.0 to detect regions where, over an average of ten windows of 100 bases and a minimum of 250 consecutive bases, the GC content is more than 50% and the calculated observed:expected ratio of the number of CpG sites is > 0.6. To expand on the set identified by *cpgplot*, we combined this information with the DNA methylation data from bisulfite sequencing. To this end, we defined hypomethylated CpG sites as positions where <50% of the reads support a methylated cytosine. We then implemented a hidden Markov model with six emission probabilities, corresponding to the four 'standard' nucleotides, as well as a hypomethylated C and a hypomethylated G—cytosine in the complementary strand—and eight hidden states (corresponding to an 'island' and 'non-island' state for each 'standard' nucleotide). Since we do not have a gold standard set of CpG islands in zebra finch, to set the transition and emission probabilities, we based ourselves on the CpG islands identified by *cpgplot*. We then used the function *MultinomialHMM* in the Python package hmmlearn (v 0.2.6) to identify positions in the zebra finch genome supporting the 'island' states [129], merging those <50 bps away from each other, and keeping merged stretches longer than 150 bps and shorter than 2 kb. By this approach, the CpG islands identified contain the vast majority of those identified by *cpgplot* plus some islands that were presumably too short or missed for other reasons.

### Detection of crossover events

To identify autosomal crossover events in the zebra finch pedigrees, we analyzed a total of 54 meioses across the four multi-sibling pedigrees (Fig 1A), and implemented a previously described algorithm to call crossovers [69]. The algorithm uses the transmission of "informative sites" (i.e., positions that are heterozygous in one parent but not in the other) to phase the haplotypes inherited by the offspring and identify crossover events in the germline of the parents. We focused on polymorphic positions where all individuals in the focal pedigree had a depth of coverage of at least eight reads, none of the individuals had an indel within 10 bps, and masked heterozygous genotypes with evidence of allelic imbalance (based on the p-value of a two-sided exact binomial test > 0.01). The detection of crossover events was based on 18,872,093 informative sites.

Briefly, for each F0 parent with multiple sequenced offspring, we considered each offspring in turn as the "template" individual. The alleles in the non-template offspring are re-coded to "1" if they match the parental allele transmitted to the template, "2" if they match the untransmitted allele, or "0" if they have a missing genotype. After this re-coding, positions at which a non-template offspring changes from copying one allele to the other are defined as a "switch" (e.g., sequences such as "..1 1 2 2.."). We called putative crossovers in the template individual as having occurred within intervals in which

the majority (in practice, usually all) of the non-template individuals with genotype calls showed a switch. In turn, for F1 individuals with sequenced offspring, we re-coded informative sites as "1" and "2" states in the offspring depending on the F0 individual they were inherited from. We then called putative crossovers as transitions from one state to another.

This procedure is susceptible to genotyping errors, e.g., missed heterozygotes in the focal individual, which generate changes of phase in close physical succession. To address this issue, we grouped putative crossovers within ten informative sites of each other in each of the template individuals, and kept only those clusters with an odd number of changes of phase, classifying events with more than a single phase change as "complex". In addition, we removed crossovers within ten informative markers of the edges of the chromosomes.

To increase the resolution of putative crossover locations, we relaxed the filtering criteria for polymorphic positions. Specifically, we identified previously masked informative sites within each putative crossover interval by reducing the minimum depth of coverage to 5X and the distance to indels to 5 bps. We redefined the crossover interval if a single phase switch occurred in the same template individual as the original crossover location. This approach improved the resolution of 439 crossovers, reducing the median interval length from 849 to 647 bps.

Finally, to identify and exclude potential false-positive crossovers, we filtered genomic regions exhibiting an unusually high density of events within individual meioses. While two closely spaced crossovers may arise from class II recombination events (see Results), clusters of three or more are less biologically plausible. Motivated by this consideration, we identified clustered events in each meiosis by grouping crossovers occurring within 2 Mb of each other. This analysis revealed 55 crossovers in 17 clusters containing three or more events. Notably, these regions co-occur across multiple meioses: after merging overlapping clusters, we identified nine independent regions spanning 6 Mb in total. This pattern is consistent with genotyping errors arising from regions in the genome assembly that are challenging to analyze with short-read sequencing data. To mitigate these issues, we masked the regions within these 6 Mb, leading to the exclusion of 86 crossovers (~7% of the total) from further analysis. The median interval length after exclusion of these events is 606 bps.

## Detection of non-crossover events

While crossover events manifest as a single change of phase among the parental haplotypes, non-crossovers are indicated by two changes of phase within a relatively short genomic distance. In most of the cases, non-crossovers involve a single SNP, making it challenging to distinguish genuine non-crossover events from sequencing and genotyping errors. To keep the number of false positives at a minimum, we compared non-crossover calls obtained from GATK with those based on two extra variant callers (bcftools (version 1.9; [117]) and freebayes (version v1.3.6; [130])).

We called variants with freebayes, using a minimum mapping quality of 30, a minimum base quality of 30, and limited the analysis to the best four alternative alleles per site (version v1.3.6; [130]). To standardize the output format of variants occurring in phase and close to each other, which are represented as haplotypes in freebayes, we used vcfallelicprimitives from vcflib suite (version 1.0.10; https://github.com/vcflib/vcflib).

We used bcftools mpileup to generate a pileup format of sequencing reads with a minimum base quality of 30 and a minimum mapping quality of 30, restricting the depth to a maximum of 150 reads. We then used bcftools call to identify variants, and bcftools filter to exclude low-quality variants with a quality score below 20.

To minimize false positives, we implemented extra stringent filtering criteria. Specifically, we restricted the detection of non-crossover events to the 30 autosomes with an average mapping coverage above 20 and an average mapping quality above 40 across all individuals (S7 Fig). Moreover, we only considered genotypes meeting all of these criteria: (i) the site(s) in the putative conversion tract are consistent with the rules of Mendelian segregation, (ii) the depth of coverage is above half and below twice the average for that chromosome in that individual, and (iii) there is no evidence of allelic imbalance in heterozygous calls of the parents or any of the siblings (based on the $p$-value of a two-sided exact binomial test $> 0.05$).

To identify non-crossovers, we followed a similar approach as for crossovers, but focused on genomic segments that included two changes of phase in the same meiosis. To minimize assembly and mapping problems, and given the small

number of meiosis considered, we further required no change of phase within the same region in another meiosis. To further avoid the challenge of calling events at repetitive loci that are collapsed in the reference assembly, we only included pairs of phase changes surrounded by at least 10 congruent informative sites on each side (i.e., informative sites that do not change phase in any individual). With these criteria, we identified 158, 398, and 384 instances with two changes of phase (i.e., putative non-crossovers) with *GATK, Bcftools and freebayes*, respectively. The numbers of informative sites used to detect these changes of phase were: 18.5 millions, 14.2 millions and 14.0 millions for *GATK, Bcftools and freebayes*, respectively.

We removed non-crossover events located within 1 kb from any other. To estimate the impact of repetitive regions that are challenging to analyze with short-read sequencing, we analyzed transmission rates to the next generation as a function of the mappability around putative non-crossover events. Specifically, for each non-crossover event, we computed the fraction of mappable bps (see Mappability section) found within the two closest non-converted informative sites. Based on this analysis, we kept events in which at least 50% of bps are mappable (S25 Fig). This approach led us to detect 121, 288, and 275 non-crossovers with *GATK, Bcftools and freebayes*, respectively.

Finally, we calculated the transmission rate of non-crossovers identified by each caller, expecting that for true calls it should be approximately 50% (assuming complete power to detect transmission). Based on this criterion, we decided to base ourselves on the non-crossovers called from *freebayes* for the rest of the analyses (S5 Table).

Of the 275 events identified using *freebayes*, three are very poorly resolved: the distance between the single polymorphic site in the putative conversion tract and the closest non-converted informative site is > 10 kb and in two cases close to 1 Mb. In all three cases, heterozygosity is extremely low in these regions. We therefore excluded them from consideration and focused our analyses on the other 272 events.

**Analysis of crossover interference**

To investigate the presence of crossover interference, we followed a similar approach to that described in "Co-localization between recombination and mutation events". Specifically, we compared the distribution of distances between the closest crossover events occurring within the same meiosis ("within") to the expected distribution derived from events in different meioses ("between") (S11 Fig).

**Overlap between recombination events and CpG islands**

To compare the two types of recombination events, we restricted our attention to crossovers that occurred on the 30 chromosomes used to call non-crossovers. Crossovers that could not be resolved in an interval of < 10 kb were filtered out, leaving 882 crossovers. We used *bedtools closest* to measure the distance between a recombination event and a genomic feature, and counted the number of crossovers or non-crossovers for which the interval edges are at most 100 bps away from the closest edge of a CpG island.

To estimate the overlap expected by chance, for each recombination event defined within an interval, we generated all the locations to which the same length interval could be randomly assigned. Specifically, we kept all the windows fulfilling the following criteria: (i) within 2.5 Mb from the edge of the original interval (ii) a GC content within 2.5% the GC content of the 100 kb surrounding the original event (iii) a fraction of mappable bps within 20% of the original interval and (iv) no overlap with the the original interval. We randomly sampled 5,000 such windows for each recombination event. For each of the 5,000 sets of simulated recombination events, we tabulated the number of events that fell within 100 bps from a CpG island.

**GC-biased gene conversion (gBGC)**

To call non-crossovers, we relied on informative sites and applied stringent filters, potentially leading us to miss additional SNPs in conversion tracts. To examine the number of SNPs in the conversion tract, we reconstructed the haplotypes of each individual in the pedigree. Specifically, for each non-crossover, we extracted all SNPs present between the two flanking non-converted informative sites from the VCF file that had not passed our filtering (see section 'Detection of

non-crossover events'). In order to identify which of the additional SNPs were co-converted with the informative site(s) and thus likely in the conversion tract, we phased the biallelic SNPs using *whatshap* (version v2.1; [131]). Out of the 275 non-crossovers, we successfully reconstructed the haplotypes of all individuals across the pedigree for 103 events.

Given that the non-crossovers identified in the pedigree with three siblings show suspiciously many pairs of changes of phase per meiosis (S7 Table), suggesting that inference of non-crossovers in small families is less reliable, we excluded that family from this analysis.

Among the 103 non-crossovers for which we reconstructed the haplotypes, 75 did not include any other SNPs in the conversion tract. To these cases, we added those for which the phasing did not succeed but where there was no other SNP than the informative site in the unfiltered VCF (n = 13), totaling 88 events with only 1 SNP. Among the events where more SNPs were added to a conversion tract, some of those might be false positives; we therefore kept only the SNPs that passed the same stringent filters applied to identify informative sites. Together with the unphased events with more than one informative site, we identified 39 multi-SNPs events, including a total of 101 SNPs. We lacked support to determine if there is a single or multiple SNPs for 146 events, which remain unclassified.

The degree of gBGC was then computed for three sets of non-crossovers: (i) all the non-crossovers, (ii) the non-crossovers for which the phasing analysis confirmed the presence of a single heterozygous position in the conversion tract, and (iii) the non-crossovers with more than one heterozygous position in the conversion tract. For each set, we analyzed the informative heterozygous sites used for the detection of phase changes, which pass all filtering criteria (see "*VCF extra filtering step*" subsection). To estimate the magnitude of gBGC, we selected all the positions carrying a Strong (G or C) and a Weak (A or T) allele, and quantified the proportion of times the Strong allele was transmitted over the Weak one.

## Conversion tract length estimation

To estimate the mean length of the gene conversion tracts, we followed the approach in Li et al, 2019 [6]. This method assumes that the conversion tract length follows an exponential distribution, such that the probability of co-conversion of two informative sites $d$ sites apart is given by $e^{-\lambda d}$ (an informative site is defined as in the section "Detection of crossover events"). For each informative site inferred to be in a conversion event, we recorded its distance to the other informative sites in a window of 5,000 bps on each side, as well as whether the site were co-converted with the focal informative site. We estimated the mean conversion tract length (i.e., $1/\lambda$) by maximum likelihood over a grid of 1 bp ranging between 1–1000. In this approach, each converted site is treated as independent, even when they may not be; the resulting likelihood is therefore a composite likelihood. To estimate uncertainty, we bootstrapped the set of informative sites 1,000 times; the central 950 values were used to estimate the 95% confidence interval. We repeated this procedure using a window of 2,000 bps instead of 5,000, and the estimated mean tract length was identical and the confidence interval very similar.

## Estimating the number of non-crossovers in a meiosis

The total number of non-crossovers can be estimated from the number of observed events, given an estimate of the power to detect non-crossovers. The power depends on the probability of having at least one informative site within the conversion tract. This probability hinges on the density of informative sites and the mean conversion tract length.

To estimate our power, we considered each chromosome in each focal individual (a founder of the pedigree) separately. As for the gBGC estimates, we excluded the family with three siblings for this analysis. We took 1M draws from an exponential distribution with parameter $1/L$, where $L$ is the mean conversion length, and placed the intervals along the chromosome at random, recording the fraction that overlapped at least one informative site. We used three values of $L$: our estimate of the mean conversion tract length (32 bps) and the lower and upper bounds on the 95% confidence intervals for L (24 and 42 bps). We considered the fraction of intervals that overlapped an informative site as our estimate of the probability of detecting a non-crossover event on that chromosome in that founder.

Assuming the rate of non-crossover events is consistent in mappable and non-mappable regions, we then estimated the total number of non-crossovers on a chromosome by dividing the observed number of events by the estimate of the power. We computed the average number across founders to produce estimates per chromosome and considered the sum for the estimates per chromatid (Fig 5).

**Identifying chromosome inversions and estimating their effect on recombination**

Three autosomal chromosomes have been previously described to harbor large inversion polymorphisms in the zebra finch: chromosome 5, chromosome 11, and chromosome 13 [83]. To estimate the effect of these inversions on local rates of recombination, we first followed the approach in Harringmeyer & Hoekstra 2022 [132] to determine, for each inversion, its coordinates along with the associated karyotype states of each individual in the pedigree dataset. Specifically, we identified genomic regions indicative of polymorphic inversions using local PCA. We used biallelic SNPs (excluding singletons) to perform local PCA with the lostruct package in R [133]. Analyses were conducted on F0 and F1 generation pedigree samples and we used 50 kb windows to compute PCA map distances for each chromosome (see run_lostruct.R script; https://github.com/petrelharp/local_pca). These distances were visualized via multidimensional scaling (MDS) with two axes.

Polymorphic inversions were expected to manifest as contiguous genomic regions with distinct population structures. We used k-means clustering (k = 2–10) in MDS space, selecting the optimal k by maximizing the silhouette score. Known inversion polymorphisms on three autosomal chromosomes showed an optimal k of 3. Inversions were identified by locating the first and last 50 kb windows of contiguous clusters, with >90% intermediate windows matching. For chromosome 13, where a single inversion or several nested inversion polymorphisms encompass nearly the entire length of the chromosome [83], we assigned putative breakpoints to the first and last 50 kb window assigned to the most distinct cluster.

Karyotype states were determined via PCA on the inverted regions, expecting three clusters: two homokaryotypes (ancestral and inverted) and one heterokaryotype. SNPs were filtered with a 1 kb physical filter and a 0.1 minor allele frequency threshold using *plink* (version v.1.9; [134]). Each inversion showed clear clustering into groups AA, AB, and BB, with AB representing putative heterokaryotypes.

## Supporting information

**S1 Fig. Example of a DNM that likely occurred during the development of the proband (B1057, central panel).** For each of the five individuals in this pedigree, the nucleotides (circles) carried by pairs of sequencing reads spanning multiple informative heterozygous sites (horizontal lines) are shown. The mutated position is indicated by gray vertical lines, and colors represent groups of reads supporting the same haplotype (excluding the mutated position). In haplotypes carrying the DNM, the mutated position is highlighted by a red rectangle. In carrier individuals, all sequencing reads are represented for the mutated position (regardless of whether they span multiple informative sites). In the first generation (parents of the proband), the mother is positioned on the left. The annotation left to the panel of the proband shows the parental haplotype carrying the DNM. Each panel indicates the number of reads supporting each allele (top side), regardless of whether they span more than one informative site.
(TIF)

**S2 Fig. Cumulative distribution of the fraction of sequencing reads supporting the mutated allele.** Cumulative distribution of the fraction of sequencing reads in the proband carrying the mutated allele, for DNMs inferred to have occurred in either parental germlines (orange) and in the early development of the proband (blue). The expected fraction of ½ for inherited variants is shown as a vertical dashed line. The p-value is obtained from a Kolmogorov-Smirnov test.
(TIF)

**S3 Fig. Proportion of each of seven substitution types.** Proportion of each substitution type in DNMs inferred from pedigree sequencing (x-axis) versus in low-frequency polymorphisms segregating in a dataset of 27 non-closely related individuals (y-axis). 95% CIs (lines) were obtained by bootstrapping 5 Mb autosomal windows 500 times.
(TIF)

**S4 Fig. Distribution of interval lengths within which a crossover is inferred to have occurred.** Intervals larger than 20,000 bps are shown in one bin.
(TIF)

**S5 Fig. Crossovers per meiosis as a function of chromosome size in bps, mappability and repeat content for each chromosome.** (A) Average number of crossovers observed per meiosis for the 39 autosomes. The X axis is on a log-scale. The blue dashed line represents the minimum number of crossovers expected per chromosome per meiosis. (B) Average number of crossovers observed per meiosis as a function of the mappability of the chromosome. (C) Average number of crossovers observed per meiosis as a function of the proportion of repeat elements in the chromosome. For each panel, the red dots indicate the chromosomes for which the average number is significantly smaller than 0.5 ($p < 0.05$, using an exact binomial test).
(TIF)

**S6 Fig. Mean recombination rate per chromosome.** The left panel shows estimates from the LD-based map; the dashed red line represents the mean recombination rate for the whole genome, including micro-chromosomes (4.68 cM/MB). The right panel shows estimates from crossovers; the red dashed line is the mean recombination rate for the whole genome is 2.28 cM/Mb. The comparison of these two plots suggests that estimates of recombination rates for chromosomes <40 Mb are unreliable, consistent with simulations [51].
(TIF)

**S7 Fig. Proportion of sites that are mappable or repeat elements as a function of chromosome size, mapping quality or coverage.** (A) Fraction of the chromosome that is mappable (see Methods) as a function of chromosome size. (B) Proportion of repeat elements per chromosome as a function of chromosome size. The proportion of repeat elements was estimated by counting the proportion of positions, for each chromosome, that is assigned as a repeat in the softmask version of the genome. (C) Proportion of the chromosome that is mappable as a function of the mean mapping quality for each chromosome. (D) Proportion of the chromosome that is mappable as a function of the mean mapping coverage for each chromosome. For each panel, the red dots are the chromosomes that have been removed from the NCO analysis. The blue dots are the chromosomes that meet our mapping quality and mapping coverage threshold (see Methods).
(TIF)

**S8 Fig. Distribution of the number of informative sites in the non-crossover conversion tracts.**
(TIF)

**S9 Fig. Cumulative recombination rates inferred from LD for macro- and micro-chromosomes.** Results are shown for the six macro-chromosomes (defined as chromosomes >40 Mb in size) in the left panel and for the 24 micro-chromosomes that meet our mapping quality and mapping coverage threshold on the right. The black line represents the cumulative recombination rate, $\rho$ (=$4N_er$), estimated with LDhelmet.
(TIF)

**S10 Fig. Chromosomal inversion identification, karyotype assignment, and consequences on crossover distribution and population recombination rate.** (A) Local PCA for three autosomal chromosomes harboring inversion polymorphisms in the zebra finch, with each point representing a 50 kb window. The distance between the relatedness structure of individual windows was evaluated using multidimensional scaling (MDS) and is represented

using the MDS1 axis. Windows colored in blue belong to outlier regions associated with the candidate inversions, with the length of each inversion represented as a horizontal blue bar at the top. (B) Consistent with the occurrence of inversion polymorphisms, PCA using variants from the entirety of each outlier region recover three distinct clusters of samples along PC1. These clusters correspond to either group of homokaryotypic (i.e., groups AA and BB) individuals at opposite ends of PC1 and a group of heterokaryotypic individuals (i.e., group AB) immediately in between. Homokaryotype group assignment (i.e., AA vs. BB) was done arbitrarily for each inversion from left to right along PC1 and is not necessarily consistent with labeling in [83]. The name of each inversion is given at the top left and the approximate coordinates of each inversion is at the top right of each panel, respectively. Points are color-coded by sex for male (blue) or female (pink). (C) Cumulative distributions of crossovers (purple and orange) and population recombination rate inferred from LD-map (black) for each autosome harboring an inversion polymorphism. Crossovers from homokaryotypic and heterokaryotypic individuals represented in purple and orange, respectively. The location of each inversion is shown in blue.
(TIF)

**S11 Fig. Distribution of distances between the closest pairs of crossover events.** Comparisons of events that occurred within the same meiosis ("within", blue) or a different meiosis ("between", orange).
(TIF)

**S12 Fig. Cumulative distribution of recombination events for macro- and micro-chromosomes, in the two sexes.** Top panel: Cumulative distribution of crossover events (females in red and males in blue), for macro-chromosomes (left) and micro-chromosomes (right). The position of the events is normalized by the size of the chromosomes. For crossovers, the $p$-values for Kolmogorov-Smirnov tests comparing the two sexes are 0.31 and 0.34 for the macro-chromosomes and the micro-chromosomes, respectively. Bottom panel: Cumulative distribution of non-crossover events (females in red and males in blue), for macro-chromosomes (left) and micro-chromosomes (right). The position of events is normalized by the size of the chromosomes. For non-crossovers, the $p$-values for Kolmogorov-Smirnov tests comparing the two sexes are 0.15 and 0.32 for the macro-chromosomes and the micro-chromosomes, respectively. For a comparison of the sex-averaged distribution on macro- vs micro-chromosomes, see S4 Table.
(TIF)

**S13 Fig. Cumulative distribution of recombination events, for three categories of chromosome sizes.** Same as in Fig 2, but considering chromosomes shorter than 20 Mb separately from those between 20 and 40 Mb and macro-chromosomes.
(TIF)

**S14 Fig. Overlap of both types of recombination events (in males) with CpG islands.** Fractions of crossovers (left) and non-crossovers (right) identified in males meioses within 100 bps of a CpG island. The vertical lines show the observed overlap. The distribution for the overlap expected by chance is shown as a histogram, obtained by randomly shuffling all the events within a 2.5 Mb window on each side of their original location, matching for the GC content and ensuring similar mappability (see Methods for details).
(TIF)

**S15 Fig. CpG island density along the zebra finch genome.** The fraction of bps within CpG islands in consecutive genomic windows of 1 Mb, for the 30 autosomes for which we identified recombination events (see Methods). Vertical gray indicates transitions between different chromosomes. Horizontal lines show the mean CpG island density for the corresponding chromosome.
(TIF)

**S16 Fig. Overlap of crossovers with CpG islands and transcription start sites (TSSs).** Fraction of crossover events overlapping (**A**) a CpG island at most 10 kb away from a TSS, (**B**) a CpG island farther than 10 kb from a TSS, (**C**) a TSS at most 10 kb away from a CpG island, and (**D**) a TSS farther than 10 kb from a CpG island. To account for the difference in width between CpG islands and TSSs, we considered that a crossover event overlaps a CpG island (A and B) and a TSS (C and D) if it is closer than 100 bps or 500 bps, respectively. The observed overlap is shown by red vertical lines. The overlap expected by chance is shown as a histogram; it was obtained by randomly shuffling all crossovers events 5,000 times within 2.5 Mb of their original location, ensuring the shuffled location had a similar GC content and mappability (see Methods). TSSs were identified from the genome annotation (GCF_003957565.2) by keeping the positions annotated as "start_codon" (n = 20,400).
(TIF)

**S17 Fig. Overlap of non-crossovers with CpG islands and TSSs.** Fraction of non-crossover events overlapping (**A**) a CpG island at most 10 kb away from a TSS, (**B**) a CpG island farther than 10 kb from a TSS, (**C**) a TSS at most 10 kb away from a CpG island, and (**D**) a TSS farther than 10 kb from a CpG island. To account for the difference in width between CpG islands and TSSs, we considered that a non-crossover event overlaps a CpG island (A and B) and a TSS (C and D) if it is closer than 100 bps or 500 bps, respectively. The observed overlap is shown by red vertical lines. The overlap expected by chance, shown as a histogram, was obtained by randomly shuffling all crossovers events 5,000 times within 2.5 Mb of their original location, ensuring the shuffled location had a similar GC content and mappability (see Methods).
(TIF)

**S18 Fig. Overlap of crossovers and non-crossovers with CpG islands or TSSs in collared flycatcher. (A)** Fraction of crossovers detected less than 100 bps from a CpG island. **(B)** Fraction of non-crossovers detected less than 100 bps from a CpG island. **(C)** Fraction of crossovers detected less than 100 bps from a TSS. **(D)** Fraction of non-crossovers detected less than 100 bps from a TSS. The vertical lines show the observed overlap. The overlaps expected by chance are shown as histograms, obtained by randomly shuffling 5,000 times all the events within a 2.5 Mb window on each side of their original location, matching for the GC content but not for the mappability (see Methods for details).
(TIF)

**S19 Fig. Cumulative distribution of distances between DNMs and the closest crossover.** In red are the distances between events that occurred in the same germline (i.e., detected in the same proband and assigned to the same parental chromosome) and in gray between events that occurred in different germlines. The p-value was obtained from a Kolmogorov-Smirnov test.
(TIF)

**S20 Fig. Mutation rate estimates obtained from pedigree sequencing in vertebrates.** The average mutation rate per generation (top) and per year (bottom) is shown for 48 species of vertebrates. Data from [32] (their S2 Table); their estimates for mammals and birds are indicated by circles, while our point estimates for zebra finch are represented by crosses.
(TIF)

**S21 Fig. Example of a region around a putative DNM with evidence of three distinct haplotypes in one of the parents (B1047, top-left panel).** See S1 Fig for more details about how to interpret the figure. Note that the proband has no sequenced descendants, and thus only three individuals (instead of five) are shown.
(TIF)

**S22 Fig. Histogram of pairwise relatedness for the 27 unrelated individuals.** The relatedness was assessed for each pair of individuals with *relatedness2* tool (VCFtools suite).
(TIF)

**S23 Fig. LD-based hotspots overlap with CpG islands.** Fractions of hotspots within 100 bps of a CpG island, within 10 kb to a TSS (left), or greater than 10 kb from a TSS (right). The vertical red lines show the observed overlap. The overlap expected by chance is shown as a blue histogram, and was obtained by randomly shuffling all the events 5,000 times within a 2.5 Mb window on each side of their original location, matching for the GC content and ensuring a similar mappability (see Methods for details).
(TIF)

**S24 Fig. Recombination events overlap with LD-based hotspots.** Fraction of recombination events within 100 bp from a LD-based hotspot. The overlap expected by chance is shown as a blue histogram, and was obtained by randomly shuffling all the events 5,000 times within a 2.5 Mb window on each side of their original location, matching for the GC content and ensuring a similar mappability (see Methods for details).
(TIF)

**S25 Fig. Transmission rate for different sets of non-crossovers.** The sets were defined depending on the fraction of the length of the non-crossover interval that is mappable in the reference genome (see Method section "Detection of non-crossover events"). The number of non-crossovers in each set is in parentheses on the x axis label. The 95% confidence interval is represented by the black bars. The dashed red line denotes the expectation of 0.5 (assuming perfect power to detect a heterozygous site).
(TIF)

**S26 Fig. Distribution of DNMs along the mappable autosomal regions.** Colors correspond to chromosomes. The p-value is for a Kolmogorov-Smirnov test against a uniform distribution.
(TIF)

**S1 Table. Sample information.** For each individual of the pedigrees or the population the table provides information, such as, the (i) the parents ID (for pedigree data), (ii) the mapping coverage, the date of hatching (for pedigree data).
(XLSX)

**S2 Table. De novo mutation and recombination events.** List of all the mutations, crossovers and non-crossovers identified in this study.
(XLSX)

**S3 Table. Number of DNMs and low frequency polymorphism alleles for each SBS signature.**
(XLSX)

**S4 Table. Numbers of recombination events identified in maternal and paternal meioses.** The p-value was obtained from an exact binomial test of no sex difference. For the crossovers, the number of maternal meioses is higher than paternal meioses, and we adjusted the null model accordingly.
(XLSX)

**S5 Table. Non-crossover transmission rates for three different variant callers.** Transmission is tested only for configurations in which the genotypes of the offspring, the partner and the grandkid allow us to track the conversion event (see Methods). Shown as CIs are the 95% confidence intervals from an exact binomial test.
(XLSX)

**S6 Table. *P*-values for the Kolmogorov-Smirnov tests presented in Fig 2.** For each set of chromosomes, we tested if the distribution of crossovers is the same as that of non-crossovers, and whether the distribution of each type of recombination events are uniform. We also compared the distribution of crossovers (non-crossovers) on macro- vs micro-chromosomes.
(XLSX)

**S7 Table. Average number of pairs of phase changes per family per meiosis.**
(XLSX)

**S1 Data. Numerical data that underlies graphs or summary statistics.**
(XLSX)

**S2 Data. Numerical data that underlies S9 Fig.**
(TXT)

**S1 Text. Command lines.**
(DOCX)

## Acknowledgments

We thank Sarah London (University of Chicago) for providing two zebra finch testis samples as well as members of the Andolfatto, Przeworski, and Sella labs for helpful discussions.

## Author contributions

**Conceptualization:** Felix Wu, Marc de Manuel, Molly Przeworski.

**Data curation:** Djivan Prentout, Marc de Manuel.

**Formal analysis:** Djivan Prentout, Daria Bykova, Marc de Manuel.

**Funding acquisition:** Marc de Manuel, Molly Przeworski.

**Investigation:** Djivan Prentout, Daria Bykova, Marc de Manuel.

**Methodology:** Djivan Prentout, Carla Hoge, Marc de Manuel, Molly Przeworski.

**Project administration:** Molly Przeworski.

**Resources:** Carla Hoge, Daniel M. Hooper, Callum S. McDiarmid, Simon C. Griffith.

**Supervision:** Marc de Manuel, Molly Przeworski.

**Validation:** Djivan Prentout.

**Visualization:** Djivan Prentout, Marc de Manuel.

**Writing – original draft:** Djivan Prentout, Marc de Manuel, Molly Przeworski.

**Writing – review & editing:** Djivan Prentout, Daria Bykova, Carla Hoge, Felix Wu, Marc de Manuel, Molly Przeworski.

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
