## [Decision Letter · Decision Letter 0]

14 Feb 2025

PGENETICS-D-24-01379

Mutation and recombination parameters in zebra nch are similar to those in mammals

PLOS Genetics

Dear Dr. Prentout,

Thank you for submitting your manuscript to PLOS Genetics. After careful consideration, we feel that it has merit but does not fully meet PLOS Genetics's publication criteria as it currently stands. Therefore, we invite you to submit a revised version of the manuscript that addresses the points raised during the review process.

The reviewers are the same ones who evaluated your submission to PLoS Biology. The reviewers are enthusiastic about the findings, and so am I. However, the emphasis on comparisons with mammals and the arguments about selection still need to be toned down. Please consider this request carefully, along with the other remaining points raised by the reviewers.

Please submit your revised manuscript within 30 days Mar 16 2025 11:59PM. If you will need more time than this to complete your revisions, please reply to this message or contact the journal office at plosgenetics@plos.org. Please include the following items when submitting your revised manuscript:

We look forward to receiving your revised manuscript.

Kind regards,

Bret Payseur

Academic Editor

PLOS Genetics

Kelly Dyer

Section Editor

PLOS Genetics

Aimée Dudley

Editor-in-Chief

PLOS Genetics

Anne Goriely

Editor-in-Chief

PLOS Genetics

**Journal Requirements:**

https://journals.plos.org/plosgenetics/s/submission-guidelines#loc-parts-of-a-submission

5) Please ensure that the funders and grant numbers match between the Financial Disclosure field and the Funding Information tab in your submission form. Note that the funders must be provided in the same order in both places as well.

**Reviewers' comments:**

Reviewer's Responses to Questions

**Comments to the Authors:**

Reviewer #1: Thank you to the authors for addressing my previous comments. I still feel positive towards the paper and find that it is improved/clearer. It’s an impressive amount of robust analysis. However, I still have some comments about the framing and conclusions of the study.

After reading and considering the comments of the other reviewers, I still think that this is not a comparative study (as stated on Line 29) and that the strong emphasis of comparing to mammals (as is done in the title, and for each statement in Lines 34 – 40) is not necessarily justified. Perhaps the authors could argue that comparisons can be invoked in terms of measures that are rare (i.e., de novo mutation). However, in terms of recombination, I think that there have been enough measures in birds and mammals over many years to show that (a) zebra finch recombination rate is probably lower than other passerine species and (b) not all mammals are hovering around 1 cM/Mb, as is clearly stated in lines 68-69 (indeed, this does not match the statement in Lines 61 & 62). The authors have acknowledged this in their response, and extensive discussions in recombination mechanisms (lines 84 – 102 – of which the authors have made a huge contribution) yet strong statements remain. Generally, I agree that comparison is important, but the *strong* emphasis on comparison with mammals detracts from interesting conclusions of this paper that stand on their own merit.

Similarly, I am still not convinced that the current manuscript offers any novel conclusions on natural selection on recombination. The authors invoke arguments related to genetic variation persisting in populations (Lines 597 – 600), but don’t spell out why this is relevant. The authors have also shown throughout the manuscript that recombination is highly variable across birds and mammals. I don’t disagree that there is likely selection occurring (indeed, selective constraints have been discussed over decades of literature summarised in various reviews, e.g. Arter & Keeney 2024, Stapley et al 2017, Bomblies & Henderson 2022, etc. etc.), but I think the authors oversell the contribution of this manuscript in answering this question.

Line 33: information “for or from” 80 meioses?

Lines 74-75: The Arter and Keeney review would support conservation of mechanisms over broad phylogenetic distances.

Lines 128 – 130: At this point of reading, I don’t know what the COSMIC mutational signatures are.

Lines 589 – 590: But the Priore and Pigozzi paper shows remarkable variation in CO count using MLH1 foci.

Reviewer #2: The manuscript represents a comprehensive analysis of the characteristics of mutation and meiotic recombination in zebra finch. I still think that this study provides important and valuable data and is useful for researchers in the fields of avian genomics and evolution. For the present version, authors carefully addressed the comments and performed additional analyses. Please find below my comments on the current version of the manuscript.

1) For the comment about stabilizing selection on recombination characteristics, I understand the authors’ arguments regarding their similarity between taxa despite millions of years of evolution. However, as the authors rightly point out, discussing the scale of RR variation between taxa is beyond the scope of this study. As well as inferring selection regimes of recombination modifiers. I think the current version (with slightly modified title, abstract and toned-down claims) better aligns with the study’s aims.

2) Following the comment about double crossovers and the potential detection of a type II non-interfering pathway of CO resolution in birds, authors analyzed their data and claimed that 8% of all 1,174 detected crossovers are located closer than 1 Mb to their neighbor. The authors also suggest that these instances may have resulted from complex resolutions with template switching.

I was interested in this data and decided to take a closer look at it.

From S3 Table, there are 126 chromatids with more than one CO (293 COs in total). Among them, 38 chromatids (30%) contain COs located closer to each other than 1 Mb (with an average distance of 277 Kb). Interestingly, they are distributed very unevenly throughout the genome. Of these 38 chromatids, 17 (45%) belong to microchromosomes from 25 to 37. If we include COs separated by a distance of 1 to 2 Mb, this percentage increases to 50%. One theoretical possibility that comes to mind is that the type II pathway may act predominantly in microchromosomes as a backup mechanism to ensure their correct segregation. However, the complete absence of such events on most medium-sized chromosomes (14-15, 17-24) seems strange.

Moreover, among the 126 chromatids, there were 20 with more than two COs located relatively close to each other (74 COs in total), with 6 of them including “complex” COs. For example, there are four cases on chromosome 3, where 3 or 5 COs are located within the distance of ~650 kb. Moreover, all four cases are located in the same chromosome region. There is also a similar region on chromosome 2 (where 3 cases of 3 COs are located within the same ~800 kb region). The same applies to 6 cases on chromosome 13, which are slightly more widely distributed but also include up to 5 crossovers.

While it is theoretically possible that high-intensity hotspot usage may be responsible for the repeated inclusion of a region in recombination, it does not explain multiple template switching. The type II pathway also does not convincingly explain this (and it is also questionable whether it plays a role, given the skewed distribution of “non-interfering” COs across the genome). Does the phenomenon of complex resolutions with template switching sufficiently explain such instansies?—How long and complex could these events be?

While hypothesising on these questions might be out of the scope of this paper, it is necessary to ensure that these data are not caused by some kind of technical and/or methodological issue and to determine whether it is reliable to consider such “complex-complex” events as 3/5 COs instead of a single event.

3) Regarding the minor comment about references ( 6) Please revise the reference list. For example, the citation of Soler et al. (2021) from p.4 is missing), the authors stated that this issue was fixed, but the current reference does not appear to match the described content—“primordial germ cell phenotypes before gonadal development” (line 83). Could you please clarify?

“Soler MA, Medagli B, Wang J, Oloketuyi S, Bajc G, Huang H, Fortuna S, de Marco A. 2021. Effect of Humanizing Mutations on the Stability of the Llama Single-Domain Variable Region. Biomolecules 11: 163”

I also have a couple of other comments.

4) line 538: the observed number of crossovers used to infer the NCO:CO ratio (Table 1) is 18.5. I assume this number results from removing nine smallest microchromosomes from the crossover dataset (as was done for non-crossovers) but I didn't find any mention of this in either Results or Methods sections (which I think would be good to have).

5) in the table S4, the number of crossover events and the p-value (the second column) do not correspond to the data in lines 289-292.

Reviewer #3: The authors have satisfactorily addressed my initial comments and have also included new analyses suggested by the other referee.

**Have all data underlying the figures and results presented in the manuscript been provided?**

Reviewer #1: Yes

Reviewer #2: Yes

Reviewer #3: Yes

PLOS authors have the option to publish the peer review history of their article (what does this mean? ). If published, this will include your full peer review and any attached files.

**Do you want your identity to be public for this peer review?** For information about this choice, including consent withdrawal, please see our Privacy Policy .

Reviewer #1: No

Reviewer #2: No

Reviewer #3: No

**Figure resubmission:**
---

## [Editor Report · Decision Letter 1]

20 Mar 2025

Dear Dr Prentout,

Thank you for resubmitting your manuscript entitled "Germline mutation rates and fine-scale recombination parameters in zebra finch" for consideration by PLOS Genetics. I appreciate the revisions you made in response to reviewer concerns. In particular, the context and motivation for the study has been substantially clarified through meaningful changes to the Introduction.

We are pleased to inform you that your manuscript has been editorially accepted for publication in PLOS Genetics. Congratulations!

Yours sincerely,

Bret Payseur

Academic Editor

PLOS Genetics

Kelly Dyer

Section Editor

PLOS Genetics

Aimée Dudley

Editor-in-Chief

PLOS Genetics

Anne Goriely

Editor-in-Chief

PLOS Genetics

Comments from the reviewers (if applicable):

**Data Deposition**

http://datadryad.org/submit?journalID=pgenetics&manu=PGENETICS-D-24-01379R1

**Press Queries**

---

## [Editor Report · Acceptance letter]

PGENETICS-D-24-01379R1

Germline mutation rates and fine-scale recombination parameters in zebra finch

Dear Dr Przeworski,

We are pleased to inform you that your manuscript entitled "Germline mutation rates and fine-scale recombination parameters in zebra finch" has been formally accepted for publication in PLOS Genetics! Your manuscript is now with our production department and you will be notified of the publication date in due course.

With kind regards,

Anita Estes

PLOS Genetics

On behalf of:
